# Anti-Nogo-A NG101 treatment induces changes in spinal cord micro- and macrostructure following spinal cord injury

Lynn Farner [1], Paulina S. Scheuren [2,3], Kiomars Sharifi [1], Tim M. Emmenegger [1,4], Maryam Seif[1,5], Michèle Hubli[1], Martin Schubert [1], Marc Bolliger [1], Rüdiger Rupp [6,7], Norbert Weidner [6,7], Rainer Abel [8], Doris Maier [9], Klaus Röhl[10], Michael Baumberger[11], Margret Hund-Georgiadis[12], Marion Saur[13], Jesús Benito[14], Kerstin Rehahn[15], Mirko Aach[16], Andreas Badke[17], Jiri Kriz[18], Tim Killeen[1], Alan J. Thompson [19], Nikolaus Weiskopf [5,20,21], Martin E. Schwab[22], Armin Curt[1], Patrick Freund [1,5,21] ✉ & the Nogo Inhibition in Spinal Cord Injury (NISCI) Study Group*

NG101 is a recombinant antibody that neutralizes the nerve growth inhibitor Nogo-A, promoting neural repair and improving upper extremity motor function in spinal cord injury (SCI). This study evaluated spinal cord MRI biomarkers to detect treatment-related structural changes and enhance patient stratification using data from 106 participants with acute cervical SCI in the phase 2b NISCI trial. We assessed lesion volume, tissue bridges, and remote changes in cross-sectional cord area (CSA), and tract-specific myelin-sensitive magnetization transfer saturation (MTsat) over six months. Compared to placebo, NG101-treated participants exhibited faster lesion volume reduction and a slower decline of CSA and MTsat in the corticospinal tracts and dorsal columns. Crucially, multimodal stratification incorporating MRI and electrophysiological measures substantially enhanced the detection of clinical treatment effects. These findings suggest NG101 slows trauma-induced progressive macro- and microstructural degeneration or promotes fiber sprouting. Combining MRI with electrophysiology enables sensitive detection of treatment effects and efficient trial designs. ClinicalTrials.gov identifier: NCT03935321.

Traumatic spinal cord injury (SCI) causes substantial and often lasting clinical impairment due to the limited regenerative capacity of the central nervous system (CNS)[1]. One promising therapeutic strategy to enhance axonal plasticity is the inhibition of Nogo-A, a potent neurite growth suppressive membrane protein in CNS myelin and neuronal membranes. Preclinical studies show that NG101, a humanized monoclonal antibody targeting Nogo-A, promotes axonal sprouting and functional recovery[2–4]. Recent exploratory findings from a phase 2b clinical trial suggest that NG101 may also improve upper extremity motor function in participants with motor incomplete cervical SCI, supporting the translational potential of Nogo-A inhibition[5].

To better understand how Nogo-A inhibition influences structural recovery in human SCI, and to sensitively track potential regenerative effects, objective in vivo biomarkers are needed[6]. Individuals with SCI vary widely in injury severity and recovery potential, complicating the detection of treatment responses. Quantitative spinal cord imaging offers a way to characterize both focal and remote structural changes,

A full list of affiliations appears at the end of the paper. *A list of authors and their affiliations appears at the end of the paper.
✉e-mail: patrick.freund@balgrist.ch

**Fig. 1 | Flowchart of participant screening, randomization, and analysis populations.** This CONSORT-style flowchart summarizes the progression of participants through the study. The diagram specifies that 463 individuals were assessed for eligibility, with 129 enrolled and randomized. Participants were assigned to either the NG101 (Verum; $n = 80$) or placebo ($n = 49$) groups. The Full Analysis Set included 78 NG101 and 48 placebo biologically independent participants. For neuroimaging, the full imaging dataset comprised 63 NG101 and 43 placebo participants, while the Multi-Parameter Mapping (MPM) subset from 8 centers included 32 NG101 and 30 placebo participants. Reasons for exclusion at each stage, such as withdrawal, lack of consent, or technical artifacts, are indicated. N indicates the number of biologically independent participants.

providing insight into the biological mechanisms of treatment and a means to enhance stratification. Tissue bridges (TB) at the injury site, especially when analyzed at the level of individual funiculi and fiber tracts, can predict functional preservation of specific descending or ascending connections[7]. Cross-sectional cord area (CSA) is a robust macroscopic marker of spinal cord atrophy, reflecting structural loss from axonal degeneration and demyelination, secondary to traumatic SCI and correlating with clinical impairment[8]. Microstructural imaging using Multiparameter Mapping (MPM)[9] provides reproducible metrics such as magnetization transfer saturation (MTsat), which is sensitive to myelin content and enables detection of demyelination and remyelination of afferent and efferent spinal fiber tracts. Both CSA and MTsat are suitable for multicenter trials and allow sensitive, longitudinal assessment of spinal cord integrity[10].

A major challenge in the context of SCI clinical trials is related to the inherently heterogeneous population in terms of injury characteristics and recovery potential. This often hinders the reliable evaluation of treatment effects and underscores the need to identify homogeneous subgroups of participants and those who are likely to respond to intervention. Objective biomarkers such as electrophysiology (e.g., somatosensory evoked potentials [SSEPs]) can reduce variability and improve statistical power in detecting treatment effects. When integrated with such electrophysiological assessments, imaging biomarkers may further enhance patient stratification by capturing preserved neural structure and function[11].

Despite these advances, no study has yet assessed how macro- and microstructural Magnetic Resonance Imaging (MRI) biomarkers respond to a targeted neuroregenerative treatment in acute cervical SCI, or how these measures can be combined to optimize trial design. In this study, we investigated whether CSA and MPM-derived indices can track NG101-induced structural effects and whether, based on preserved tissue bridges, patient stratification can potentially be improved in future trials. We analyzed longitudinal changes in spinal cord structure at and above the lesion site and explored the combined utility of MRI and electrophysiological measures to identify responders and enhance the efficiency of future trials.

## Results

### Participants

Between May 20, 2019, and July 20, 2022, 463 individuals with acute traumatic cervical SCI were screened for eligibility in the Nogo Inhibition in Spinal Cord Injury (NISCI) trial (ClinicalTrials.gov identifier: NCT03935321). Of these, 140 met all inclusion criteria and consented. Following withdrawals before ($n = 11$) and after ($n = 2$) randomization, and one exclusion due to spontaneous recovery prior to the baseline visit assessment 126 participants were enrolled (NG101: $n = 78$; placebo: $n = 48$, mean age $46.2 \pm 16.7$ years; 84.9% male, American Spinal Injury Association Impairment Scale (AIS) grades placebo: A 22.9%, B 25.0%, C 33.3%, D 18.8%; AIS grades NG101: A 33.4%, B 17.9%, C 26.9%, D 21.8%). MRI data were available for 106 participants, after excluding those who declined consent for MRI data analysis ($n = 12$), did not undergo MRI for medical reasons ($n = 1$), organizational issues ($n = 3$) or scans which were excluded due to artefacts ($n = 4$) (Fig. 1). This cohort (mean age $47.5 \pm 16.3$ years, 84.1% male) comprised 63

participants randomized to NG101 and 43 to placebo. The distribution of AIS grades at baseline was similar between groups (AIS grades placebo: A 20.9%, B 23.3%, C 34.9%, D 20.9%; AIS grades NG101: A 23.8%, B 23.8%, C 28.6%, D 23.8%).

Analyses were conducted based on data availability, therefore sample sizes varied between analyses. Analyses of lesion parameters (tissue bridges, lesion volume, lesion length, lesion width, CST and DC damage) were performed on the full imaging dataset (NG101: $n = 63$; placebo: $n = 43$, 84.1% male). Due to unsuccessful automated registration to the anatomical PAM50 template, analyses of CST and DC damage were performed on a final cohort of 60 participants in the NG101 group and 35 in the placebo group (87.4% male). Stratification by tissue bridge status, determined from screening-visit data, was available for 59 NG101 and 39 placebo participants (82.7% male), as not all participants completed all imaging sessions. CSA and MTsat values were derived from MTsat maps obtained via the MPM protocol, which was not implemented at all sites due to scanner field strength limitations. MPM data were acquired at eight centers for 39 NG101 and 31 placebo participants (mean age 45.8 ± 16.4 years; 87.1% male; AIS grades placebo: A 16.1%, B 25.8%, C 35.1%, D 22.6%; AIS grades NG101: A 25.6%, B 33.3%, C 23.1%, D 17.9%). Following exclusions due to inadequate image quality, CSA and MTsat analyses were conducted in a subset of 62 participants, including 32 from the NG101 group and 30 from the placebo group (mean age 45.9 ± 16.4 years; 90.3% male; AIS grades placebo: A 16.7%, B 23.3%, C 33.3%, D 26.7%; AIS grades NG101: A 28.1%, B 18.8%, C 28.1%, D 25.0%). Due to unsuccessful automated registration to the anatomical template PAM50, analyses of MTsat values in CST and DC tracts were performed on a final cohort of 25 participants in the NG101 group and 22 in the placebo group (91.5% male).

Tibial SSEPs were successfully recorded in 74 NG101 and 41 placebo participants (mean age 45.7 ± 16.8 years; 86.1% male; AIS grades placebo: A 22.0%, B 24.4%, C 34.1%, D 19.5%; AIS grades NG101: A 33.8%, B 17.6%, C 28.4%, D 20.3%); C8 dSSEPs were obtained in 73 NG101 and 41 placebo participants (mean age 45.9 ± 16.8 year; 86% male; AIS grades placebo: A 22.0%, B 26.8%, C 31.7%, D 19.5%; AIS grades NG101: A 32.9%, B 17.8%, C 28.8%, D 20.5%).

Nine participants were missing both Upper Extremity Motor Score (UEMS) and Spinal Cord Independence Measure (SCIM) self-care data at follow-up (NG101: $n = 7$; placebo: $n = 2$), and two additional participants were missing UEMS data only (NG101: $n = 1$; placebo: $n = 1$). Consequently, UEMS follow-up data were available for 70 NG101 and 45 placebo participants (mean age 45.9 ± 16.9 years; 85.2% male; AIS grades placebo: A 22.2%, B 24.4%, C 35.6%, D 17.8%; AIS grades NG101: A 35.7%, B 15.7%, C 27.1%, D 21.4%), and SCIM self-care data for 71 NG101 and 46 placebo participants (mean age 46.1 ± 16.8 years; 83.3% male; AIS grades placebo: A 21.7%, B 26.1%, C 34.8%, D 17.4%; AIS grades NG101: A 34.2%, B 15.5%, C 28.2%, D 21.1%).

### Macrostructural findings

At the screening visit, NG101-treated participants had significantly larger lesion volumes than those receiving placebo (mean ± SD: 394.1 ± 233.4 mm³; 699.8 ± 872.1 mm³; p = 0.016, t(64.76) = −2.47, Cohens d = −0.44 [95%CI −0.86 to −0.02], 95% CI of the difference = −552.40 to −58.91). Lesion length was also greater in the NG101 group (18.78 ± 16.55 mm; 14.49 ± 8.28 mm; $p = 0.093$, t(90.45) = 1.70, Cohen's $d = 0.31$ [95% CI −0.10 to 0.72], 95% CI of the difference = −0.73 to 9.32), while lesion width was comparable between groups (5.81 ± 2.15 mm; 5.64 ± 1.83 mm; $p = 0.68$, t(89.75) = 0.42, Cohen's $d = 0.08$ [95% CI −0.32 to 0.49], 95% CI of the difference = −0.64 to 0.97). Over time, lesion volume declined more rapidly in the NG101 group (−52.14 mm³/month, 95%CI −78.96 to −25.33) than in the placebo group (−2.25 mm³/month, 95%CI −33.46 to 28.96; t(98.3) = 2.45, $p = 0.01$, $\eta_p^2 = 0.06$, 95% CI of the difference 9.48 to 90.30) (Supplementary Fig. 1). A similar pattern was observed for lesion length, which

decreased more in NG101-treated participants (−1.03 mm/month, 95% CI −2.38 to 0.31) than in placebo (−0.39 mm/month, 95%CI −0.98 to 0.19; $p = 0.10$, t(167.99) = 1.58, $\eta_p^2 = 0.02$, 95% CI of the difference = −1.45 to 0.16). In contrast, lesion width declined at similar rates in both groups (NG101: −0.19 mm/month, 95%CI −0.46 to 0.09; placebo: −0.16 mm/month, 95%CI −0.28 to −0.04; $p = 0.73$, t(126.83) = 0.33, $\eta_p^2 = 0.001$, 95% CI of the difference = −0.19 to 0.14). Corticospinal Tract (CST) damage decreased more rapidly in the NG101 (−1.00 %/month, 95%CI −1.94 to −0.07) compared to the placebo group (0.58 %/month, 95%CI −0.60 to 1.76, $p = 0.04$, $t(108.74) = 1.99$, $\eta_p^2 = 0.05$, 95% CI of the difference = −3.16 to −0.01). Dorsal Column (DC) damage declined at similar rates in the placebo group (−0.15 %/month 95%CI −1.15 to 0.86), and in the NG101 group (−0.66 %/month 95%CI −1.45 to 0.13, $p = 0.41$, $t(145.67) = 0.76$, $\eta_p^2 = 0.01$, 95% CI of the difference = −1.86 to 0.83) (Supplementary Fig. 2). Midsagittal tissue bridges did not differ between groups at the screening visit (NG101: 1.35 ± 1.75 mm; placebo: 1.40 ± 1.31 mm; $p = 0.87$, t(94.55) = 0.17, Cohen's $d = −0.03$ [95% CI −0.44 to 0.37], 95% CI of the difference = −0.67 to 0.56) and remained stable over time in both groups (NG101: +0.06 mm/month, 95%CI −0.10 to 0.21; placebo: + 0.08 mm/month, 95%CI 0.01 to 0.15; $p = 0.61$, $t(121.18) = 0.06$, $\eta_p^2 = 0.00001$, 95% CI of the difference = −0.13 to 0.12).

Above the level of lesion (C1-C2 level), at the screening visit, CSA was similar between treatment groups (NG101: 69.86 ± 6.46 mm²; placebo: 66.25 ± 6.03 mm²; $p = 0.065$, t(40.77) = 1.89, Cohens $d = 0.58$ [95%CI −0.04 to 1.18], 95%CI of the difference −0.24 to 7.46), but declined significantly faster in placebo-treated participants (−0.60 mm²/month, 95%CI −0.88 to −0.32) than in those receiving NG101 (−0.01 mm²/month, 95% CI −0.32 to 0.17; t(70.3) = 2.53 $p = 0.007$, $\eta_p^2 = 0.13$, 95% CI of the difference 0.11 to 0.95) (Fig. 2).

At the same level, anterior-posterior width of the spinal cord was similar between treatment groups at the screening visit (NG101: 7.87 ± 0.43 mm; placebo: 7.84 ± 0.59 mm; $p = 0.85$, t(34.52) = 0.19, Cohen's $d = 0.06$ [95% CI −0.54 to 0.66], 95% CI of the difference = −0.29 to 0.35), but declined significantly faster in placebo-treated participants (−0.05 mm/month, 95%CI −0.08 to −0.02) than in those receiving NG101 (0.01 mm/month, 95%CI −0.01 to 0.04; $p = 0.002$, t(75.1) = 2.90, $\eta_p^2 = 0.20$, 95% CI of the difference = 0.02 to 0.11). Left-right width of the spinal cord was different between treatment groups at the screening visit (NG101: 11.22 ± 0.72 mm; placebo: 10.70 ± 0.63 mm; $p = 0.015$, t(41.00) = 2.55, Cohen's $d = 0.77$ [95% CI 0.15 to 1.39], 95% CI of the difference = 0.11 to 0.94), but declined at a similar rate in placebo-treated participants (−0.03 mm/month, 95%CI −0.06 to 0.008) and those receiving NG101 (−0.03 mm/month, 95% CI −0.06 to- 0.001; $p = 0.83$, t(69.2) = 0.20, $p = 0.846$, $\eta_p^2 = 0.0008$, 95% CI of the difference = −0.05 to 0.04) (Supplementary Fig. 3).

### Microstructural findings

In the CST at level C1-C2, MTsat declined faster in the placebo group (−0.09 units/month, 95%CI −0.15 to −0.03) than in the NG101 group (0.02 units/month, 95%CI −0.04 to 0.07; $p = 0.020$, t(46.10) = 2.14, $\eta_p^2 = 0.26$, 95% CI of the difference = 0.01 to 0.21). In the DC, MTsat also declined faster in the placebo group (−0.20 units/month, 95%CI −0.25 to −0.14) than in those receiving NG101 (−0.12 units/month, 95%CI −0.16 to −0.07; $p = 0.026$, t(74.63) = 2.12, $\eta_p^2 = 0.09$, 95% CI of the difference = 0.005 to 0.16) (Supplementary Fig. 4).

### Stratification with MRI and electrophysiology

As reported previously[5], participants with motor-incomplete SCI who received NG101 showed increased UEMS recovery (time:treatment group interaction, $p = 0.009$, t(242.62) = 2.63, mean ΔUEMS placebo: 2.39 [95% CI 1.44-3.35, mean ΔUEMS NG101: 3.17 [95% CI 2.79-3.55]) compared to the placebo group with a moderate effect size. Post-hoc power analyses revealed an estimated required sample size ($N_{Req}$) of 82 to achieve statistical significance with 80% power in this subgroup of

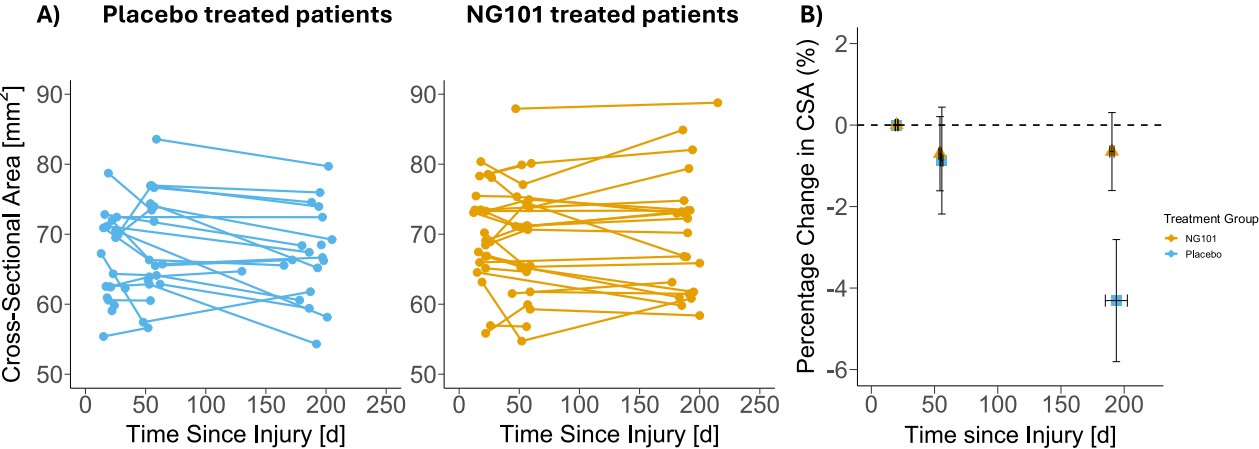

**Fig. 2 | Longitudinal changes in cervical spinal cord cross-sectional area (CSA). A** Individual trajectories of absolute CSA at the C1-C2 level over time (days since injury) for biologically independent placebo-treated (blue circles, *n* = 30) and NG101-treated (orange circles, *n* = 32) participants. **B** Percentage change in CSA over time relative to the baseline screening visit. Data are presented as mean values ± standard error of the mean (SEM) for both the percentage change in CSA (vertical error bars) and the time since injury (horizontal error bars) for the NG101 (orange triangles) and placebo (blue squares) groups. Total *n* = 62 biologically independent participants (NG101: *n* = 32, Placebo: *n* = 30). Source data are provided as a Source Data file.

participants with motor-incomplete SCI (Table 1). SCIM self-care recovery was larger for the NG101 group (*p* = 0.0071, t(182.46) = 2.72, mean ΔSCIM placebo: 1.33 [95% CI 0.53-2.13], mean ΔSCIM NG101: 2.01 [95% CI 1.69-2.32]) compared to the placebo group with a small effect size necessitating a sample size of 120 participants with motor-incomplete SCI to detect meaningful differences (Fig. 3, Supplementary Fig. 5, 6).

In contrast, stratification based on a combination of preserved midsagittal tissue bridges (TB ≥ 1.0 mm) and the presence of tibial SSEP markedly improved treatment effect sizes. In this subgroup, there was a higher UEMS recovery for the NG101 group (*p* = 0.044, t(77.95) = 2.06, placebo: 2.15 [95% CI 0.22-4.08], NG101: 3.33 [95% CI 2.53-4.13]) with a large effect size, reducing the required sample size to just 32 participants. In addition, higher recovery of SCIM self-care was observed for the NG101 group compared to placebo (*p* = 0.000074, t(59.12) = 4.29, placebo: 1.10 [95% CI − 0.29-2.50], NG101: 2.89 [95% CI 2.31-3.46]) with effect sizes even more pronounced, requiring a minimal sample size of only 10 participants. Among participants with preserved tibial SSEP but smaller TB (< 1.0 mm), no difference in recovery of UEMS and SCIM self-care over time was observed between NG101 and placebo.

A similar pattern emerged when stratifying by C8 dSSEP in combination with TB thickness. Participants with both preserved C8 dSSEP and TB ≥ 1.0 mm in the NG101 group showed greater recovery in both UEMS (*p* = 0.021, t(95.33) = 2.34, placebo: 2.3 [95% CI 0.64-3.96], NG101: 3.46 [95% CI 2.78-4.15]) and SCIM self-care (*p* = 0.013, t(70.98) = 2.55, placebo: 1.38 [95% CI 0.08-2.68], NG101: 2.37 [95% CI 1.84-2.91]) compared to the placebo group with large effect sizes corresponding to required sample sizes of 42 and 32 participants, respectively. Among participants with smaller TB (< 1.0 mm), no differences in clinical recovery were observed between treatment groups.

The NISCI trial enrolled 126 participants over 39 months using UEMS as a stratifier. Enriching this inclusion criteria with preserved tibial SSEP and tissue bridges ≥1.0 mm the estimated enrollment period was shortened to 27 months (−31%) (Supplementary Table 1).

## Discussion

MRI biomarkers provide an effective means to assess structural changes and potentially monitor therapeutic effects in SCI[6]. This study demonstrates that quantitative spinal cord MRI can sensitively detect both focal and remote structural changes rostral to the lesion in response to NG101 treatment[5]. Our findings suggest that NG101 promotes tissue preservation and attenuates degeneration along sensorimotor pathways, with potential regenerative effects (even in a cohort including motor-complete participants) as demonstrated by slower progression of CSA atrophy. Stratification based on SSEP reduced the required sample size to detect a significantly higher UEMS recovery in NG101 compared to placebo participants to 66, compared to 82 participants required with clinical stratification alone[11]. MRI-based biomarkers further enhanced patient stratification: combining spared tissue bridges (TB ≥ 1.0 mm) with preserved tibial SSEP yielded a large effect size (Cohen's *d* = 0.92) for UEMS recovery in the NISCI participants, reducing the required sample size further to 32 participants. These results underscore the utility of MRI and electrophysiology as complementary tools for improving sensitivity to treatment effects and more efficiently guiding stratified clinical trial designs.

NG101-treated participants showed significantly greater reductions in lesion volume over time than placebo-treated participants, despite having larger lesions at the screening visit. Despite this imbalance, lesion width and midsagittal tissue bridges were comparable between treatment groups, and only lesion length was greater in the NG101 group, indicating similar preserved spinal fiber tracts at the screening visit. This effect was not due to a difference in timing of MRI assessments, as earlier MRI lesions are typically larger due to edema[7,12]. The steeper decline in lesion volume in the NG101 group may partly reflect early resolution of edema or hemorrhage, but could also involve axonal sprouting, formation of new connections, or glial reorganization in the perilesional area[13]. Notably, the faster apparent volume reduction was primarily observed within the CST, whereas the DC did not show a comparable decline. This tract-specific pattern suggests that the observed effect is not due to generalized Wallerian degeneration, which would also affect the DC, but may instead reflect remodeling or plasticity within descending motor pathways. Preclinical evidence supports that Nogo-A inhibition promotes axonal plasticity and rewiring around the lesion[2]. Thus, the lesion-volume trajectory may reflect a combination of spontaneous fluid resolution and potential drug-induced tissue effects, although the relative contribution of each cannot be definitively determined from the current data. In contrast, lesion volumes remained relatively stable over time in placebo-treated participants.

Remote treatment effects were also evident as the spinal cord area at C1-C2 declined markedly in the placebo group but remained stable in NG101-treated participants. Previous studies in traumatic SCI have shown spinal cord degeneration patterns like those observed in the

**Table 1 | Effect sizes and estimated sample sizes for NG101 versus placebo comparisons across stratification strategies**

| | | Delta UEMS | | | | | | | Delta SCIM self-care | | | | | | |
|---|---|---|---|---|---|---|---|---|---|---|---|---|---|---|---|
| | | N | Mean | SD | Mean Difference (Verum – Placebo) [95% Bootstrap CI] | Effect size (Cohen's d) [95% Bootstrap CI] | Required total sample size for 80% power (Nreq) | Achieved Power (%) | N | Mean | SD | Mean Difference (Verum – Placebo) [95% Bootstrap CI] | Effect size (Cohen's d) [95% Bootstrap CI] | Required total sample size for 80% power (Nreq) | Achieved Power (%) |
| **I. Clinical Stratification** | | | | | | | | | | | | | | | |
| **Injury Severity** | | | | | | | | | | | | | | | |
| Motor-Complete | NG101 | 36 | 8.28 | 5.90 | -1.39 [-4.93,2.16] | 0.22 [-0.80,0.35] | 530 | 19 | 36 | 4.44 | 4.75 | -2.01 [-4.45,0.58] | 0.42[-1.02,0.11] | 146 | 45 |
| | Placebo | 21 | 9.67 | 6.90 | | | | | 22 | 6.45 | 4.91 | | | | |
| Motor-Incomplete | NG101 | 34 | 19.74 | 8.15 | 4.20 [0.38, 8.04] | 0.56 [0.05,1.11] | 82 | 66 | 35 | 11.14 | 7.16 | 3.06 [-0.38,6.42] | 0.46 [-0.05,1.04] | 120 | 53 |
| | Placebo | 24 | 15.54 | 6.85 | | | | | 24 | 8.08 | 6.15 | | | | |
| **II. Electrophysiological Stratification** | | | | | | | | | | | | | | | |
| **Tibial SSEPs** | | | | | | | | | | | | | | | |
| Abolished | NG101 | 45 | 11.76 | 8.08 | 0.09 [-4.10,4.34] | 0.01 [-0.53,0.55] | 207068 | 5 | 46 | 5.11 | 5.30 | -0.98 [-3.58,1.61] | 0.19 [-0.73,0.31] | 690 | 18 |
| | Placebo | 21 | 11.67 | 8.19 | | | | | 22 | 6.09 | 5.01 | | | | |
| Preserved | NG101 | 21 | 19.48 | 9.19 | 5.24 [-0.01,10.47] | 0.63 [0.01,1.33] | 66 | 59 | 21 | 14.05 | 5.87 | 5.46 [1.68,9.15] | 0.92 [0.27,1.86] | 32 | 87 |
| | Placebo | 17 | 14.24 | 7.49 | | | | | 17 | 8.59 | 6.01 | | | | |
| **II. Electrophysiological Stratification** | | | | | | | | | | | | | | | |
| **c8 dSSEPs** | | | | | | | | | | | | | | | |
| Abolished | NG101 | 39 | 11.64 | 8.01 | 0.90 [-3.41,5.10] | 0.12 [-0.44,0.68] | 1824 | 11 | 40 | 5.20 | 6.26 | -1.35 [-4.29,1.56] | 0.24 [-0.83,0.28] | 436 | 22 |
| | Placebo | 19 | 10.74 | 7.50 | | | | | 20 | 6.55 | 4.99 | | | | |
| Preserved | NG101 | 26 | 18.46 | 9.26 | 3.83 [-1.06,8.72] | 0.45 [-0.13,1.05] | 126 | 42 | 26 | 12.12 | 5.81 | 4.33 [0.86,7.76] | 0.73 [0.15,1.44] | 50 | 77 |
| | Placebo | 19 | 14.63 | 7.84 | | | | | 19 | 7.79 | 6.07 | | | | |
| **III. Imaging Stratification** | | | | | | | | | | | | | | | |
| **Midsagittal Tissue Bridges** | | | | | | | | | | | | | | | |
| <1.0 mm | NG101 | 31 | 13.52 | 8.89 | 2.76 [-1.82,7.43] | 0.34 [-0.25,0.92] | 220 | 29 | 32 | 7.19 | 6.50 | -0.46 [-3.49,2.62] | 0.08 [-0.71,0.47] | 3642 | 8 |
| | Placebo | 17 | 10.76 | 7.34 | | | | | 17 | 7.65 | 4.46 | | | | |
| ≥1.0 mm | NG101 | 21 | 17.10 | 9.40 | 2.48 [-2.58, 7.53] | 0.30 [-0.33,0.91] | 286 | 24 | 21 | 9.57 | 7.57 | 2.00 [-2.21, 6.20] | 0.28 [-0.31,0.96] | 310 | 23 |
| | Placebo | 21 | 14.62 | 7.24 | | | | | 21 | 7.57 | 6.47 | | | | |
| **IV. Ephys (preserved tibial SSEP) + Imaging Stratification** | | | | | | | | | | | | | | | |
| **Midsagittal Tissue Bridges** | | | | | | | | | | | | | | | |
| <1.0 mm | NG101 | 9 | 18.11 | 9.10 | 4.40 [-2.53,11.84] | 0.57 [-0.41,1.74] | 78 | 29 | 9 | 12.67 | 6.38 | 3.24 [-2.01,8.33] | 0.58 [-3.40,2.09] | 76 | 29 |
| | Placebo | 7 | 13.71 | 5.99 | | | | | 7 | 9.43 | 4.58 | | | | |
| ≥1.0 mm | NG101 | 9 | 22.00 | 9.73 | 8.44 [0.10,16.90] | 0.92 [0.01,2.25] | 32 | 59 | 9 | 16.22 | 2.95 | 9.33 [4.70,13.85] | 1.86 [1.00,3.99] | 10 | 98 |
| | Placebo | 9 | 13.56 | 8.52 | | | | | 9 | 6.89 | 6.45 | | | | |

**Table 1 (continued) | Effect sizes and estimated sample sizes for NG101 versus placebo comparisons across stratification strategies**

| | | Delta UEMS | | | | | | | Delta SCIM self-care | | | | | |
|---|---|---|---|---|---|---|---|---|---|---|---|---|---|---|
| | | N | Mean | SD | Mean Difference (Verum – Placebo) [95% Bootstrap CI] | Effect size (Cohen's d) [95% Bootstrap CI] | Required total sample size for 80% power (Nreq) | Achieved Power (%) | N | Mean | SD | Mean Difference (Verum – Placebo) [95% Bootstrap CI] | Effect size (Cohen's d) [95% Bootstrap CI] | Required total sample size for 80% power (Nreq) | Achieved Power (%) |
| IV. Ephys (preserved C8 dSSEP) + Imaging Stratification | | | | | | | | | | | | | | | |
| Midsagittal Tissue Bridges | | | | | | | | | | | | | | | |
| <1.0 mm | NG101 | 12 | 17.17 | 9.66 | 3.50 [−4.91,12.39] | 0.39 [−0.69,1.53] | 166 | 18 | 12 | 11.25 | 6.06 | 4.92 [−0.43,10.00] | 0.88 [−0.09,2.29] | 34 | 51 |
| | Placebo | 6 | 13.67 | 8.31 | | | | | 6 | 6.33 | 5.13 | | | | |
| ≥1.0 mm | NG101 | 11 | 21.55 | 8.73 | 6.65 [−0.12,13.67] | 0.80 [−0.05,1.84] | 42 | 55 | 11 | 13.73 | 5.57 | 5.33 [0.29,10.21] | 0.90 [0.05,2.18] | 32 | 63 |
| | Placebo | 10 | 14.90 | 7.91 | | | | | 10 | 8.40 | 6.24 | | | | |

Recovery in Upper Extremity Motor Score (UEMS) and SCIM self-care subscore at 6 months post-baseline (Δ6-months - baseline) is shown for five stratification approaches: (I) clinical stratification (motor-complete vs. motor-incomplete), (II) electrophysiological stratification using preserved vs. absent tibial SSEP or C8 dermatomal SSEP (dSSEP), (III) MRI-based stratification using midsagittal tissue bridge (TB) thickness (<1.0 mm vs. ≥ 1.0 mm), (IV) combined MRI and tibial SSEP stratification, and (V) combined MRI and C8 dSSEP stratification. Cohen's d effect sizes with 95% confidence intervals are reported and classified as negligible (d < 0.20), small (0.20 ≤ d < 0.50), medium (0.50 ≤ d < 0.80), or large (≥0.80). Mean differences (Verum - Placebo) in recovery scores (Δ6-months - baseline) with 95% confidence intervals are also shown. Estimated total sample sizes (Nreq) indicate the number of participants required to detect a treatment effect with 80% power (α = 0.05). Higher Cohen's d values, larger mean differences, and smaller Nreq indicate more efficient stratification. Confidence intervals and hypothesis tests are two-sided and uncorrected for multiple comparisons across subgroups.

placebo group[14–16]. The observed alterations of the spinal cord area are likely linked to widespread demyelination and axonal degeneration of the axotomized ascending tracts[17,18]. These changes are primarily attributed to progressive anterograde and retrograde degeneration[17,19]. In addition, secondary pathological mechanisms such as excitotoxicity, ischemia, and inflammation may contribute to further tissue damage and hinder recovery following SCI[13]. The decelerated decline in remote spinal cord area observed in the NG101 groups suggests that treatment with anti-Nogo-A antibodies may mitigate secondary degeneration rostral to the lesion, potentially through regenerative sprouting and regeneration of injured axons, as well as compensatory sprouting of intact fiber tracts[2,4]. Similar degeneration patterns were observed in anterior-posterior width, which declined in the placebo group but was preserved in NG101-treated participants, likely reflecting protection of dorsal column pathways, which are otherwise strongly affected by Wallerian degeneration[17]. Left-right width was higher at the screening visit in the treatment group, but rates of decline were equally small in the NG101 and placebo groups, potentially reflecting retrograde changes in the descending tracts and suggesting a slower progression[20].

Quantitative myelin imaging revealed significant longitudinal decreases in MTsat in both the CST and DC, consistent with ongoing demyelination after SCI[15,20]. However, the placebo group exhibited significantly faster rates of MTsat loss, particularly in the CST, while the NG101 group showed relative stability. These results align with prior work suggesting that MTsat is a sensitive marker of microstructural integrity and may reflect NG101-induced myelin preservation. These findings support animal research that NG101 has both neuroprotective and regenerative potential, not only limiting ongoing damage, but possibly promoting axonal sprouting or structural plasticity in spared fibers[2–4].

Furthermore, structural preservation with NG101 was also observed in motor-complete participants, e.g., AIS A and B, even though no significant improvements in UEMS or SCIM self-care were detected in this subgroup[5]. This finding is consistent with the proposed regenerative effects of anti-Nogo-A antibodies as demonstrated in animal models and suggests that potentially meaningful biological changes may occur even in the absence of significant functional recovery. Such changes could reflect enhanced axonal sprouting or preservation that remains subclinical but could support residual functions above the injury site and future interventions by providing preserved neuronal substrates for regeneration. Conversely, the absence of functional gains in motor-complete participants shows the limitations of behavioral outcomes in detecting changes related to treatment and highlights the added value of imaging biomarkers.

Our results demonstrate that refining stratification strategies can substantially enhance the ability to detect treatment effects in SCI trials. While NG101 treatment led to improvements in UEMS and SCIM self-care scores among motor-incomplete participants, stratification based on combined structural (TB) and functional (tibial SSEP or C8 dSEEPS) biomarkers measured at the screening visit revealed much larger treatment effects and markedly reduced the required sample sizes. Specifically, participants with both preserved TB and SSEPs showed the most pronounced functional gains, suggesting that these markers together qualify functionality of tissue bridges and thus may capture a synergistic profile of recovery potential. In contrast, participants with TB < 1 mm or abolished SSEP showed no treatment effect, and TB > 1 mm alone was insufficient to confer a treatment benefit without preserved SSEP. This multimodal stratification identified a responder subgroup that conventional clinical-based classification would have overlooked, highlighting that combined imaging and electrophysiological markers provide a more precise and clinically meaningful framework for predicting treatment response than clinical grouping alone. Our findings further demonstrate that it is the combination of structural integrity and functional preservation that offers

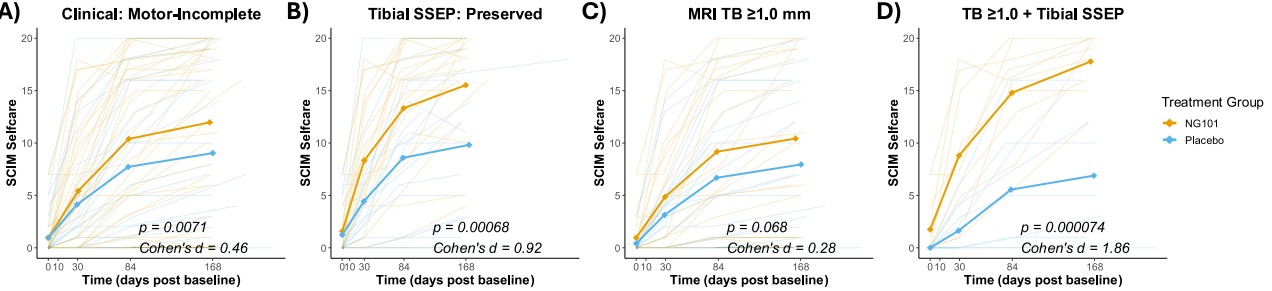

**Fig. 3 | Longitudinal recovery of SCIM self-care across clinical, electro-physiological, and imaging stratifications.** Recovery of the Spinal Cord Independence Measure (SCIM) self-care subscore is shown over time (days post-baseline) for Verum-treated (orange) and placebo-treated (blue) participants. Subgroups include: **A** Motor-incomplete SCI (Verum: $n = 35$; Placebo: $n = 24$), (**B**) Preserved tibial SSEP (Verum: $n = 21$; Placebo: $n = 17$), (**C**) Preserved midsagittal tissue bridges $\geq 1.0$ mm (Verum: $n = 21$; Placebo: $n = 21$), and (**D**) Combined preserved midsagittal tissue bridges $\geq 1.0$ mm and preserved tibial SSEP (Verum: $n = 9$; Placebo: $n = 9$). n represents biologically independent participants. Data are

presented as individual patient trajectories (transparent lines), and group estimated marginal means (solid lines with diamond markers). Statistical significance of the longitudinal trajectories (treatment-by-time interaction) was assessed using a linear mixed-effects model (LMM). Hypothesis tests were two-sided, and no adjustments were made for multiple comparisons across subgroups. Exact test statistics for the treatment contrasts are as follows: (**A**) $t(182.46) = 2.72$, $p = 0.0071$; (**B**) $t(118.16) = 3.49$, $p = 0.00068$; (**C**) $t(136.30) = 1.84$, $p = 0.068$; (**D**) $t(59.12) = 4.29$, $p = 0.000074$. Source data are provided as a Source Data file.

a robust and reliable stratification tool. While MRI alone lacked sufficient predictive precision, electrophysiological stratification based on preserved tibial SSEP proved especially valuable[11]. The combination of both electrophysiological and MRI markers further improved stratification. This concept aligns with previous work emphasizing that an integrated structural-functional perspective is essential to capture neural plasticity and repair mechanisms after spinal cord injury[21]. Such a multimodal approach may not only serve to increase statistical power and treatment effect but also to enhance trial efficiency by reducing required sample sizes. This may eventually enable more targeted and effective clinical trials in acute SCI.

Several limitations warrant consideration. This was a multi-site longitudinal study, and scanner-related variability may have introduced additional noise. Although a harmonized MPM protocol and established correction methods were applied[10,22], residual site-related effects cannot be entirely excluded, and analyses were restricted to C1-C2 due to poor image quality at C3.

While MTsat is sensitive to myelin, it is not entirely specific and may also reflect other microstructural processes[23].

Although the overall cohort was relatively large for a spinal cord imaging study, stratified subgroups were small, and sample sizes varied across analyses due to missing or excluded data, which may affect comparability.

Baseline imbalances in lesion volume within the MRI subcohort suggest that the analyzed imaging sample may not fully represent the randomized NISCI population. Although statistical adjustments were made to account for these differences, such imbalances could still influence treatment effect estimates. This limitation should be considered when interpreting the results, and future studies with larger imaging datasets will be needed to confirm these findings.

The stratification strategy was developed retrospectively in post-hoc analyses. While this approach has been validated in previous studies[11] and can provide useful estimates, its retrospective nature limits the interpretation of prospective data and carries an inherent risk of type I error. These findings should therefore be confirmed in a future clinical trial designed prospectively with the "responder" subgroup as the primary population of interest. To mitigate potential bias and strengthen the robustness of our findings, non-parametric bootstrapping (10,000 resamples) was applied to effect sizes and mean differences. Independent, prospectively designed cohorts will be required to confirm the reproducibility of these results.

In conclusion, our data suggest that NG101 antibody treatment demonstrates both focal and remote structural preservation

following cervical SCI, consistent with regenerative mechanisms seen in preclinical models[2,4]. These changes are detectable even in participants without overt functional improvement, underscoring the value of quantitative MRI in revealing treatment effects that may be missed by clinical scores alone. Stratification based on preserved structural (e.g., TB $\geq 1.0$ mm) and functional (e.g., SSEP) integrity markedly increases sensitivity to treatment effects, enhancing statistical power while reducing required sample sizes. These findings offer insights into the biological effects of NG101 on spinal cord recovery and highlight the utility of combining quantitative MRI and electrophysiology to guide more efficient, targeted, and responsive clinical trial designs in SCI.

## Methods
### Study design and participants

We analyzed spinal cord Magnetic Resonance Imaging (MRI) data collected as part of the multicenter, multinational phase 2b Nogo Inhibition in Spinal Cord Injury (NISCI) clinical trial (NCT03935321), conducted from May 2019 to July 2022. The NISCI trial received ethical approval from the relevant committees for each site: Cantonal Ethics Commission Zurich (Switzerland, 2016-01042), Heidelberg University Hospital (Germany, Afmu-815/2018), Fundació Unió Catalana Hospitals (Spain, 2016-001227-31), and University Hospital Motol (Czechia, EK-1367/19). Eligible participants were aged 18–70 years with acute traumatic cervical spinal cord injury (SCI) (neurological level of injury [NLI] C1-C8), sustained 4 to 28 days prior to the screening visit. Inclusion was based on predicted upper extremity motor scores (UEMS) at 6 months post-injury using the unbiased recursive partitioning (URP) method[5]. Participants were randomized to NG101 or placebo using a centralized system (initially 1:1, later adjusted to 3:1) to achieve a final 2:1 allocation. Both participants and study staff were blinded to treatment. Written informed consent was obtained from all participants before enrollment. Participants did not receive any compensation. NG101 or placebo was administered intrathecally at the L3-L4 level by trained clinicians. Treatment consisted of six intrathecal injections spaced 5 days apart over a 30-day period, beginning 4–28 days after injury. The NG101 group received 45 mg of the recombinant human anti-Nogo-A antibody in 3 mL phosphate-buffered saline (generated by Wyss Zurich Translational Medicine Platform, drug substance ATI355 antibody was provided by Novartis Pharma, Basel, Switzerland); the placebo group received 3 mL phosphate-buffered saline only.

All participants received standard-of-care acute surgical management and rehabilitation at their respective centers[5]. Inclusion and

exclusion criteria are described in the Supplementary Information. Primary outcomes of the NISCI study were published previously[5], this work focuses on secondary outcomes.

## MRI Data acquisition and processing

MRI data were collected at three visits: screening (mean $17.2 \pm 7.1$ days post-injury, 2–28 days prior to intervention), 1 month ($53.3 \pm 6.9$ days), and 6 months ($190.1 \pm 21.3$ days). All scans were acquired on 1.5 or 3 Tesla MRI systems. Scanner details are provided in Supplementary Table 2. Lesion volume and midsagittal tissue bridges (TB) width were quantified from sagittal T2-weighted MRI scans[24]. The midsagittal slice was defined as the central slice at the injury epicenter showing the maximal extent of preserved tissue. Manual delineation of lesion boundaries was based on T2w hyper-intensity and was done on each sagittal slice using JIM7 (Xinapse Systems) by an operator blinded to the timing of the MRI scan and treatment group (Supplementary Fig. 7). Lesion volumes were calculated by summing the lesion areas across slices and multiplying by the slice thickness (2–4.4 mm). Corticospinal tract (CST) and dorsal column (DC) damage was assessed by determining the overlap between segmented lesion volume and identified tracts in the axial plane. Participants were grouped by TB ($\geq 1.0$ mm vs $<1.0$ mm), with the cutoff derived from analyses of a multicenter cohort using a URP model of 3-month UEMS outcomes[7].

Quantitative myelin-sensitive imaging was performed using a Multiparameter Mapping (MPM) protocol that included magnetization transfer saturation (MTsat) weighted, proton density-weighted, and T1-weighted scans. Scanner harmonization was ensured by following a standardized multicenter protocol, previously validated[10,22], which demonstrated consistent parameter estimates in the cervical spinal cord across seven scanners, with intra- and inter-site coefficients of variation ranging from 2.5–12% for MT, R1, and PD, and 1.1-4.0% for morphometric measures. These images were used to compute quantitative maps of MTsat via the hMRI toolbox (version 0.2.0)[25] embedded in SPM12 (UCL, London, UK)(version 7219), which applies corrections based on separately acquired $B1^+$ and $B1^-$ maps to account for transmit and receive field inhomogeneities. Due to poor image quality at C3 across the imaged cohort, analysis focused on the C1-C2 levels, which was above the lesion level for all participants. Processing was performed using the Spinal Cord Toolbox (SCT)(version 7.0)[26] with an automated pipeline for registration, warping, and the extraction of both morphometric and microstructural parameters. Spinal cord segmentation was performed using the *sct_deepseg* function, and the resulting masks were visually inspected and manually corrected when needed using FSL(version 1.12.3). The corrected masks were registered to the PAM50 template[27] using a combination of affine and nonlinear transformations, and reverse deformation fields were applied to warp the white matter (WM) and gray matter (GM) atlases into subject space. To mitigate partial volume effects, cross-sectional area (CSA) and tract-specific metrics were extracted using SCT's atlas-based weighted masks and maximum a posteriori estimation, which account for mixed tissue contributions at voxel boundaries. Representative axial slices from each study center are shown in Supplementary Fig. 8.

## Clinical assessments

The Upper Extremity Motor Score (UEMS), the primary endpoint of the NISCI trial, is part of the International Standards for Neurological Classification of Spinal Cord Injury (ISNCSCI) and reflects motor function in the upper limbs. It is based on the manual testing of five key muscles on each side, resulting in a total score range of 0 to 50. The Spinal Cord Independence Measure (SCIM), a secondary endpoint, evaluates functional independence across daily life activities. For this analysis, only the SCIM self-care subscore (0 to 20), which assesses tasks such as feeding, dressing, and hygiene, was used to specifically evaluate upper limb functional recovery. Both UEMS and SCIM were

assessed at baseline (one day prior to intervention), and at 1, 3, and 6 months post-baseline.

Participants were grouped according to injury severity at screening as motor-complete (AIS A-B) or motor-incomplete (AIS C-D), based on the American Spinal Injury Association Impairment Scale (AIS) grade.

## Electrophysiological assessments

All participants underwent electrophysiological testing at the screening visit according to a protocol established within the European Multicenter Study about Spinal Cord Injury (EMSCI)[11]. Bilateral tibial somatosensory evoked potentials (SSEPs) and C8 dermatomal somatosensory evoked potentials (dSSEPs) were recorded in response to repetitive electrical stimulation of tibial nerves and C8 dermatomes (palmar fifth digit), respectively. SSEPs were classified as absent (amplitude below noise level defined as $0 \mu V$) or preserved(amplitude above noise level defined as $>0 \mu V$) by experienced examiners. SSEP recordings from the better side (the side with the largest amplitude) were used for further analyses. Tibial SSEPs and C8 dSSEPs were evaluated independently; for a participant's data to be included in further analyses, at least one side (right or left) of either the tibial SSEP or the C8 dSSEP had to be preserved. If SSEPs were only recorded from one side, and this side was absent, then the response for this participant was classified as not available/NA.

## Statistical analysis

All statistical analyses were conducted using R (version 4.3.1). All frequentist inferential statistics are reported as the test statistic with corresponding degrees of freedom, p-values, effect size statistics (Cohen's d, partial eta squared $\eta_p^2$), and 95% confidence intervals (CI) of the difference. Between-group differences at the screening visit were tested using independent-sample t-tests using the t.test() function in R, with effect sizes reported as Cohen's d. Longitudinal changes in MRI biomarkers were assessed using linear mixed-effects models using the lmer() function of the lme4()(version 1.1-35.1) package in R. Fixed effects included treatment group (NG101 vs. placebo), time (days post baseline), their interaction (time × treatment), time to treatment initiation, age, sex, AIS, NLI, and the baseline value of the outcome measure. Center was modeled as a random intercept to account for site-related differences when model convergence was achieved; otherwise, it was included as a fixed effect. Subject-specific random slopes for time were included to capture within-subject changes over time. The estimated rates of change for each group reflect the average linear change in the outcome over the entire follow-up period (baseline to 6 months). Group differences were assessed through the interaction between treatment group and time. Stratification strategies were evaluated based on: (1) clinical stratification: motor-complete vs. motor-incomplete, (2) electrophysiological biomarkers: preserved vs. absent tibial SSEP or C8 dSSEP, (3) MRI-based biomarkers: TB $\geq 1.0$ mm vs. $<1.0$ mm, (4) combined stratification: integration of preserved TB ($\geq 1.0$ mm) and preserved tibial SSEP or C8 dSSEP.

Stratification findings were based exclusively on data acquired at the screening visit. For every stratification strategy, longitudinal changes in a) UEMS and b) SCIM self-care were assessed using linear mixed-effects models using the lmer() function of the lme4() package in R with treatment group (NG101 vs. placebo) and time as fixed effects and a random intercept for each patient to account for within-subject variability. Group differences were assessed through the interaction between treatment group and time.

Effect sizes (Cohen's d) for changes in UEMS and SCIM self-care scores recovery ($\Delta$[6months-baseline]) between treatment groups (NG101 vs. placebo) were calculated and interpreted as negligible ($d < 0.20$), small ($0.20 \geq d < 0.50$), medium ($0.50 \geq d < 0.80$), and large ($\geq 0.80$). Required sample sizes for each stratification method to detect treatment effects with 80% power ($\alpha = 0.05$) were estimated

based on observed effect sizes. To provide bias-corrected effect estimates, we applied non-parametric bootstrapping (10,000 resamples) to calculate both effect sizes and mean differences between NG101 and placebo, together with their 95% percentile confidence intervals. Enrollment feasibility was assessed using real-world screening and inclusion rates from the NISCI trial.

To assess structure-function relationships, Pearson correlation coefficients were calculated between baseline and changes (Δ[6months-baseline]) in MRI biomarkers (e.g., lesion metrics, CSA, MTsat) and 6-month functional recovery measures (UEMS and SCIM self-care). Significant correlations are reported in Supplementary Table 3. All results are presented with effect estimates, p-values, and 95% confidence intervals.

### Reporting summary

Further information on research design is available in the Nature Portfolio Reporting Summary linked to this article.

### Data availability

Source data for all figures and tables are provided with this paper. Remaining deidentified participant data underlying the results reported in this article are not publicly available due to patient privacy and ethical restrictions. However, these data will be made available to qualified investigators upon reasonable request to the corresponding author. Access is subject to approval by the Nogo Inhibition in Spinal Cord Injury steering board and will be available beginning 12 months and ending 36 months after publication Source data are provided in this paper.

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

### Acknowledgements

We thank all participants, their families and carers for their time and commitment to this study, as well as all collaborating radiologists, radiographers and clinicians at the clinical and research sites across this multicenter study for their help and contribution to the study. We thank Novartis Pharma (Basel, Switzerland) for the generous provision of the anti-Nogo-A antibody drug substance (ATI355). This investigator-initiated trial received primary funding through the European Union's Horizon 2020 research and innovation program (grant agreement No. 681094-NISCI) and the Swiss State Secretariat for Education, Research and Innovation. Additional support was provided by Wings for Life (Salzburg), the Swiss Paraplegic Foundation, and the CeNeReg project of Wyss Zurich (University of Zurich and ETH Zurich). N. Weiskopf was supported by the Deutsche Forschungsgemeinschaft (DFG, German Research Foundation; Project No. 347592254 [WE 5046/4-2]), the Federal Ministry of Education and Research (BMBF; support code 01ED2210), and the BMBF (01EW1711A & B) within the ERA-NET NEURON framework.

## Author contributions

L.F. and P.F. directly accessed and verified the underlying raw data. L.F. and P.S.S. contributed to the conception, methodology, and analysis of the manuscript; M.Se., A.C., and P.F. were responsible for data acquisition; L.F. drafted the original manuscript; L.F., P.S.S., K.S., T.M.E., M.Se., M.H., M.Sch., M.Bo., R.R., N.Weid., R.A., D.M., K.Rö., M.Ba., M.H.G., M.Sa., J.B., K.Re., M.A., A.B., J.K., T.K., A.J.T., N.Weis., M.E.S., A.C., and P.F. contributed to the review and editing of the manuscript. Members of the NISCI Study Group contributed to data collection and site management

## Funding

## Competing interests

N. Weidner, R. Abel, D. Maier, M. Saur, K. Röhl, K. Rehahn, M. Baumberger, M. Hund-Georgiadis, J. Benito, M. Aach, A. Badke, J. Kriz, N. Weiskopf, P. Freund, and A. Curt received funding through their respective institutions via the European Commission (Grant Agreement 681094) or Wings for Life for the conduct of the work presented in this manuscript. R. Rupp received funding from the European Multicenter Study about Spinal Cord Injury (EMSCI) and serves as the unpaid Chair of the International Standards Committee and Board Member of the American Spinal Injury Association (ASIA). A. Curt is a member of the Scientific Advisory Board of the Wings for Life Foundation (Salzburg, Austria) and the International Foundation for Research in Paraplegia (Zurich, Switzerland). N. Weiskopf is supported by the Swiss State Secretariat for Education, Research and Innovation (contract 15.0137). He holds a patent on the acquisition of MRI data during spoiler gradients (US 10,401,453 B2) and has received grants from the German Federal Ministry for Education and Research (BMBF), the German Research Foundation (DFG), and the Max Planck Society (MPS). He participates in the steering committee of the NIH BRAIN Initiative U24 grant (NexGen) and chairs the Scientific Advisory Board of the Leibniz Institute for Neurobiology. His employers, the Max Planck Institute for Human Cognitive and Brain Sciences and Wellcome Center for Human Neuroimaging, have institutional research agreements with Siemens Healthcare. All other authors declare no competing interests.

## Additional information

[1]Spinal Cord Injury Center, Balgrist University Hospital, University of Zurich, Zurich, Switzerland. [2]International Collaboration on Repair Discoveries, University of British Columbia, Vancouver, Canada. [3]Department of Anesthesiology, Pharmacology, and Therapeutics, Faculty of Medicine, University of British Columbia, Vancouver, BC, Canada. [4]Department of Biomedical Imaging and Image-Guided Therapy, High Field MR Center, Medical University of Vienna, Vienna, Austria. [5]Department of Neurophysics, Max Planck Institute for Human Cognitive and Brain Sciences, Leipzig, Germany. [6]Medical Faculty Heidelberg, Heidelberg University, Heidelberg, Germany. [7]Spinal Cord Injury Center, Heidelberg University Hospital, Heidelberg, Germany. [8]Clinic for Paraplegia, Klinikum Bayreuth GmbH, Bayreuth, Germany. [9]Spinal Cord Injuries, Berufsgenossenschaftliche Unfallklinik Murnau, Murnau, Germany. [10]BG Klinikum Bergmannstrost Halle gGmbH, Zentrum für Rückenmarkverletzte und Klinik für Orthopädie, Halle, Germany. [11]Swiss Paraplegic Centre, Nottwil, Switzerland. [12]Clinic of Neurorehabilitation and Paraplegiology, REHAB Basel, 4055 Basel, Switzerland. [13]Orthopädische Klinik, Hessisch Lichtenau, Germany. [14]Fundacio Institut d'Investigacio en Ciencies de la Salut Germans Trias i Pujol, Badalona, Barcelona, Spain. [15]Treatment Centre for Spinal Cord Injuries, Trauma Hospital Berlin, Berlin, Germany. [16]Department of Spinal Cord Injuries, BG University Hospital Bergmannsheil, Bochum, Germany. [17]BG Clinic Tübingen, Tübingen, Germany. [18]Spinal Cord Unit, Department of Rehabilitation and Sports Medicine, 2nd Faculty of Medicine, Charles University and University Hospital Motol, Prague, Czechia. [19]Queen Square MS Centre, UCL Queen Square Institute of Neurology, Faculty of Brain Sciences, UCL, London, United Kingdom. [20]Felix Bloch Institute for Solid State Physics, Faculty of Physics and Earth System Sciences, Leipzig University, Leipzig, Germany. [21]Department of Imaging Neuroscience, UCL Institute of Neurology, University College London, London, UK. [22]Institute for Regenerative Medicine, University of Zurich, Switzerland; Wyss Zurich Translational Center, University and ETH Zurich, Zurich, Switzerland. ✉e-mail: patrick.freund@balgrist.ch

## the Nogo Inhibition in Spinal Cord Injury (NISCI) Study Group

Lynn Farner [1], Paulina S. Scheuren [2,3], Maryam Seif[1,5], Michèle Hubli[1], Martin Schubert [1], Marc Bolliger [1], Rüdiger Rupp [6,7], Norbert Weidner [6,7], Rainer Abel [8], Doris Maier [9], Klaus Röhl[10], Michael Baumberger[11], Margret Hund-Georgiadis[12], Marion Saur[13], Jesús Benito[14], Kerstin Rehahn[15], Mirko Aach[16], Andreas Badke[17], Jiri Kriz[18], Tim Killeen[1], Nikolaus Weiskopf[5,20,21], Martin E. Schwab[22], Armin Curt[1] & Patrick Freund[1,5,21]

