## [Transparent Peer Review file · Nature Communications]

Anti-Nogo-A NG101 treatment induces changes in spinal cord micro- and macrostructure following spinal cord injury

Corresponding Author: Professor Patrick Freund

Version 0:

Reviewer comments:

Reviewer #1

(Remarks to the Author)

In the current manuscript, the authors investigated whether cross-sectional area (CSA) and Multiparameter Mapping (MPM)-derived indices can track NG101-induced structural effects and whether, based on preserved tissue bridges, patient stratification can potentially be improved in future trials. The authors analyzed longitudinal changes in spinal cord structure at and above the lesion site starting from screening (pre-treatment) to 1 month and then 6 months. Finally, the authors explored the combined utility of MRI and electrophysiological measures to identify responders and enhance the efficiency of future trials.

The authors have provided a strong manuscript. This reviewer would like to point out the following:

-There is a figure in the manuscript (unlabelled so it is hard to tell what number the figure is) which tracks “CSA” as y axis and the time since injury as x axis. This figure shows the preservation of CSA among cases treated with NG-101 compared to those treated with placebo. This corresponds to the lines 178-182 in the manuscript: “Above the level of lesion (C1-C2 level), at the screening visit, CSA was similar between treatment groups (NG101: $n = 32$, $69.86 \pm 6.46\text{mm}^2$; placebo: $n=30$, $66.25 \pm 6.03\text{mm}^2$; $p=0.065$), but declined significantly faster in placebo-treated participants ($-0.77\text{mm}^2/\text{month}$, 181 95%CI -1.10 to -0.44) than in those receiving NG101 ($-0.06\text{mm}^2/\text{month}$, 95% CI -0.82 to 0.70 ; $p=0.0020$).

Therefore, it is fair to assume that these CSA measurements were performed at the C1-C2 level for all patients. As such, it might be helpful to then see what level the injury occurred for each patient so that meaningful comparisons may be performed. In other words, was the level of injury accounted for in these comparisons?

-Moreover, it would also be important to include snapshots of SCT -derived segmentations to determine if there were any artifact-related issues.

-Along the same lines, the authors state in the discussion: “Our findings suggest that NG101 promotes tissue preservation and attenuates degeneration along sensorimotor pathways, with potential regenerative effects (even in a cohort including motor-complete participants) as demonstrated by slower progression of CSA atrophy.” Looking at the same figure, some subjects in both groups had a decrease in CSA from pre-treatment to 1 month and then an increase in the CSA by 6 months. While it is certainly evident from the figure that the CSA was preserved more in the patients receiving NG-101, the authors’ argument that there may be some “potential regenerative effects” attributable to NG-101 is challenged by this finding given that even some subjects in placebo demonstrate this effect.

Reviewer #2

(Remarks to the Author)

This study was a multicenter, multinational, randomized, double-blind, placebo-controlled Phase 2b clinical trial—a framework that represents the gold standard for clinical investigation—to test whether anti-Nogo-A (NG101) treatment is associated with structural preservation/repair detectable by quantitative spinal cord MRI and whether MRI+SSEP can improve trial stratification. Each of these design elements serves a critical purpose in minimizing bias and ensuring the reliability of the results. The multicenter approach, involving specialized SCI centers across Germany, Switzerland, Spain, and the Czech Republic, enhances the diversity of the patient population and improves the generalizability of the findings beyond a single institution. Randomization, which allocated participants to either the NG101 or placebo group, is essential

for preventing selection bias and creating groups that are, on average, comparable at the start of the study. The double-blinding procedure, where neither the participants nor the clinical staff knew who was receiving the active drug, is crucial for preventing performance and detection biases in both the administration of care and the assessment of outcomes. Finally, the inclusion of a placebo control group allows for the true effect of the drug to be isolated from the natural.

Major

1. Baseline imbalances and modeling choices

- Despite these considerable methodological strengths, the study suffers from a critical and ultimately unavoidable flaw that significantly impacts the interpretation of a subset of its findings. A detailed examination of the baseline characteristics reveals a statistically significant difference in a key prognostic variable between the two randomized groups. At the screening visit, before any intervention was administered, the group of participants randomized to receive NG101 had, on average, significantly larger lesion volumes than the group randomized to receive the placebo. The mean lesion volume in the NG101 group was reported as 729.1 mm³, compared to just 400.3 mm³ in the placebo group (p=0.011). This represents an 82% larger average lesion size in the active treatment arm at the outset of the study.
- This baseline imbalance is a major methodological weakness. While randomization is designed to produce comparable groups, by chance, it can sometimes fail to do so, particularly with smaller sample sizes. The failure to achieve comparability for a primary measure of injury severity introduces a powerful potential confounder. This confounder does not invalidate the entire study, but it does demand an extremely cautious and critical interpretation of any results related to changes in lesion volume over time.
- The implications of this baseline difference are profound. A primary finding reported by Farner et al. is a "faster monthly reduction in lesion volume" in the NG101 group, which is interpreted as a positive therapeutic effect indicative of drug-induced tissue remodeling or resolution of edema. However, this interpretation fails to adequately account for the initial disparity. In the acute and subacute phases of SCI, a substantial portion of the hyperintense signal that constitutes the "lesion volume" on a T2-weighted MRI scan is not necrotic tissue but rather vasogenic edema and hemorrhage^{1,2}. This fluid component resolves naturally over the first weeks and months post-injury, leading to a spontaneous reduction in the overall measured lesion volume. A larger initial lesion, particularly one with a greater inflammatory and edematous component, possesses a mathematically greater absolute volume to lose during this natural resolution process. Consequently, the observed "faster reduction" in the NG101 group may be, in whole or in significant part, an artifact of this baseline difference rather than a direct consequence of a regenerative drug effect. The authors' attribution of this finding to "axonal sprouting" or "glial reorganization" is highly speculative and potentially invalid due to this unaddressed confound. This forces a re-evaluation of the paper's evidence, shifting the burden of proof for a biological effect of NG101 away from the confounded focal lesion data and onto the analysis of remote biomarkers that are geographically distinct from the primary injury.
- The analysis of structural changes at the epicenter of the injury presents a complex and ultimately ambiguous picture. The data show a significantly faster rate of decline in both lesion volume (-19.9 mm³/month in the NG101 group vs. -2.8 mm³/month in the placebo group; p=0.005) and lesion length in the NG101-treated participants.¹ As established in the preceding section; these findings must be interpreted with extreme caution. Due to the significant baseline imbalance, where the NG101 group started with much larger lesions, these results cannot be unambiguously attributed to a therapeutic effect. It is equally plausible that they reflect the natural history of the resolution of larger initial injuries. Therefore, the evidence for a drug-induced effect on the primary lesion itself is weak and confounded.
- Consider adding fixed covariates (age, AIS grade, NLI, time-to-treatment, site) and random slopes for time to better capture inter-individual trajectories and site heterogeneity.
- The MRI subset exhibits baseline imbalances that warrant additional adjustment and sensitivity analyses. Rather than "topping up" enrollment post hoc to erase these differences—which is rarely appropriate after randomization and may introduce temporal/site biases—please (i) predefine and report ANCOVA/mixed-effects models that include key baseline MRI covariates (e.g., lesion volume/length/width, TB, time from injury), (ii) adjust or stratify by site/scanner where feasible, and (iii) provide sensitivity analyses, consider adding fixed covariates (age, AIS grade, NLI, time-to-treatment, site) and random slopes for time to better capture inter-individual trajectories and site heterogeneity. Given the mid-trial change in allocation ratio (initially 1:1, later 3:1 to achieve ~2:1 overall), please also discuss whether this operational change could have contributed to the observed baseline imbalances.

2. Unacknowledged Limitations

Beyond the authors' own assessment, a deeper critique reveals other critical issues that temper the study's conclusions.

- The Overarching Confound of Baseline Lesion Volume: This must be revisited and framed as the most significant unacknowledged weakness of the study. The failure of randomization to create comparable groups on this key prognostic variable fundamentally undermines the conclusions that can be drawn about any treatment effects on the focal lesion itself.
- The Post-Hoc Nature of the Stratification Analysis: It is crucial to stress that the powerful stratification strategy was developed through a retrospective, post-hoc analysis of the trial data. This is a valid and extremely useful approach for generating new hypotheses. However, the results do not carry the same weight of evidence as a prospectively defined analysis. The true test of this strategy will require it to be applied prospectively in a new clinical trial where the "responder" subgroup is the primary population of interest from the outset. Moreover, similar study has reported this³.

Minor:

1. Imaging protocol clarity (potential inconsistency)

- The Results/Participants section attributes limited MPM availability to "scanner field strength limitations," yet Methods state all scans were acquired on 3 T systems. This reads as contradictory. Please reconcile and detail per-site vendors/sequences, harmonization steps, and how site effects were modeled.

2. Multiplicity and statistical transparency

- Numerous outcomes are tested (lesion volume/length/width, CST/DC overlap, CSA and widths, MTsat in CST/DC, several

stratifications). No correction for multiplicity is described. Please prespecify primary imaging endpoints and apply a multiplicity control (e.g., FDR). Report effect sizes with 95% CIs consistently for all primary/secondary outcomes.

3. Stratification & “post-hoc power”

- The improvement in effect sizes using $TB \geq 1.0$ mm plus preserved SSEP is promising, but appears post-hoc and uses observed effects to back-calculate required sample sizes. This can be optimistic. Please (i) clarify whether these stratifications were pre-specified, (ii) provide bias-corrected effect sizes (e.g., bootstrap, nested CV), and (iii) avoid “post-hoc power”; instead report CIs and, if appropriate, prospective simulations for future trials.

4. Microstructural metrics

- Please add quality control details for MTsat (e.g., motion rejection thresholds, B1 correction), test–retest reliability references for cord MTsat, and discuss partial-volume handling at C1/C2. Also, clarify whether CST/DC ROIs were lesion- unaffected at C1/C2 across all subjects.

5. Manual segmentations & reproducibility

- Lesion segmentation was manual with blinded operators. Please report inter-/intra-rater reliability and make annotation guidelines/code available.

6. Define specialized terms earlier and correct terminology

- Several domain-specific measures appear in the Results without prior plain-language definitions, which may hinder readability for non-clinician readers (e.g., UEMS, SCIM, MTsat). While UEMS and the SCIM self-care subscore are described in the Methods, these definitions come after their first substantive use in the text. Please define each acronym on first mention in the Abstract/Results and consider adding a brief glossary.

7. Concomitant treatments beyond NG101

- The Intervention/Methods section clearly describes the blinded intrathecal dosing schedule of NG101 versus saline placebo, but the manuscript does not specify permitted/prohibited concomitant therapies or standardization of rehabilitation/acute surgical care across sites within this MRI study. Clarifying these co-interventions is essential to assess potential confounding of imaging and clinical outcomes.

- Please add (i) a concise description of allowed and disallowed concomitant treatments (e.g., timing of decompression/fixation, corticosteroid use, rehabilitation intensity/dose), (ii) whether these were balanced between groups, and (iii) whether any such variables were adjusted for in the longitudinal models.

Reference:

1. Margetis, K., Das, J. M. & Emmady, P. D. Spinal Cord Injuries. in StatPearls (StatPearls Publishing, Treasure Island (FL), 2025).
2. Zhang, Y. et al. Acute spinal cord injury: Pathophysiology and pharmacological intervention (Review). *Molecular Medicine Reports* 23, 1–18 (2021).
3. Pfyffer, D. et al. Prognostic value of tissue bridges in cervical spinal cord injury: a longitudinal, multicentre, retrospective cohort study. *The Lancet Neurology* 23, 816–825 (2024).

Reviewer #3

(Remarks to the Author)

Reviewer #4

(Remarks to the Author)

Key Results

This multicenter phase 2b study investigates the effects of anti–Nogo-A antibody NG101 in acute cervical SCI using quantitative MRI and electrophysiology. The authors report that NG101 treatment attenuates lesion volume growth, preserves spinal cord cross-sectional area (CSA), and slows decline in magnetization transfer saturation (MTsat) in corticospinal and dorsal column tracts. Integration of MRI biomarkers with electrophysiology (SSEP) enhanced stratification and improved effect sizes for functional outcomes (UEMS, SCIM self-care). Findings suggest NG101 may mitigate macro- and microstructural degeneration and that multimodal stratification can optimize trial design.

Validity

The study is robust in design and employs rigorous imaging and electrophysiological protocols across multiple centers. Statistical methods (linear mixed-effects modeling, effect size estimation, sample size projection) are appropriate. However, several limitations and potential sources of bias require clearer acknowledgment:

- Attrition and missing data: Imaging subsets are substantially smaller than the full cohort (e.g., CSA/MTsat analyses included only ~62 participants). If dropouts or exclusions disproportionately involved patients with more severe injuries, results could be biased toward “healthier” participants.
- Stratification: The refined stratification ($TB \geq 1$ mm + preserved SSEP) is highly informative but was conducted retrospectively, in small subgroups. This increases the risk of type I error and inflated effect sizes.
- Imaging acquisition: Advanced MPM protocols were only implemented at select sites, and scanner variability may introduce noise despite harmonization efforts.
- Structure–function disconnect: Structural preservation was observed even in motor-complete participants without functional gains, raising the possibility that some imaging changes reflect subclinical processes (e.g., edema resolution, gliosis) rather than functional regeneration.

These issues do not undermine the core findings but should temper interpretation and be more explicitly discussed.

Significance

This work is highly significant for SCI research:

- Demonstrates, for the first time, that anti-Nogo-A treatment produces measurable in vivo macro- and microstructural changes.
- Supports MRI and electrophysiology as biomarkers to improve trial sensitivity, reduce sample sizes, and refine inclusion criteria.
- Findings have broad translational relevance for neuroregeneration trials beyond SCI.

Potential limitation: the absence of clear clinical improvement in motor-complete participants may temper enthusiasm for immediate clinical impact.

Data and Methodology

Imaging and electrophysiology protocols are state-of-the-art, with careful quality control and blinded analyses. MRI biomarkers (CSA, MTsat) are well validated for multicenter studies. However, the paper should more explicitly address:

- The impact of missing data and variable sample sizes across analyses.
- How harmonization across scanners was achieved and validated.
- The rationale for selecting TB ≥ 1 mm as the cut-off, and whether sensitivity analyses using other thresholds were tested.

Analytical Approach

- Linear mixed-effects models and effect size/sample size estimations are appropriate.
- Post-hoc subgroup analyses are informative, though risk of type I error inflation should be acknowledged.
- Consider sensitivity analyses to confirm robustness of treatment effects in stratified groups.

Suggested Improvements

- Include a CONSORT-style flow diagram summarizing patient screening, randomization, exclusions, and follow-up across imaging and clinical analyses to enhance transparency.
- Clarify that stratification was based on baseline TB/SSEP status, not longitudinal changes, and that functional improvements were concentrated in those with preserved tissue and electrophysiology.
- Explicitly state that non-responders (TB < 1 mm or absent SSEP) showed no treatment effect, to avoid misinterpretation.
- Provide greater discussion of limitations: missing data, subgroup size, baseline imbalances, and potential scanner-related variability.
- Where possible, report correlations between structural biomarkers (e.g., TB, CSA, MTsat) and functional recovery (UEMS/SCIM) to strengthen biological interpretation.

Clarity and Context

- Manuscript is well-written, with clear figures and logical flow.
- Contextualization with prior work is strong (e.g., citing MRI biomarkers, Nogo-A preclinical data).
- Consider shortening certain result subsections for readability, as the manuscript is data-dense.
- The authors should emphasize the conceptual advance that multimodal stratification is superior to conventional AIS-based grouping more prominently in the Discussion.

References

The references are comprehensive and appropriate, with inclusion of both preclinical and clinical studies. Some citations to under-review work should be updated if possible.

Reviewer Expertise:

I am experienced in clinical trial design, neurorehabilitation, and imaging biomarkers in SCI. I defer detailed assessment of raw MRI sequence implementation to technical experts.

Overall Assessment:

This is a technically rigorous and conceptually innovative study. It advances the field by demonstrating that NG101 induces measurable structural preservation and that multimodal biomarker stratification can enhance trial sensitivity. With clearer acknowledgment of limitations, addition of a CONSORT-style flow diagram, and stronger emphasis on the stratification findings, the manuscript will make a valuable contribution to Nature Communications.

Version 1:

Reviewer comments:

Reviewer #1

(Remarks to the Author)

All previous comments have been adequately addressed.

Reviewer #2

(Remarks to the Author)

The authors have submitted a substantially revised manuscript that addresses the majority of the concerns raised in the previous round of review. I commend the detailed and thoughtful nature of the rebuttal.

The inclusion of the flow diagram significantly enhances the transparency of participant selection and exclusion across the clinical and imaging sub-cohorts. Furthermore, the methodological shift from post-hoc power calculations to non-parametric bootstrapping for effect size estimation represents a distinct improvement, providing more robust and bias-corrected confidence intervals. The additional sensitivity analyses presented in the rebuttal, which tested the robustness of the findings against various model specifications and covariates, are also well-received.

However, I retain a degree of skepticism regarding the baseline imbalance in lesion volume. While I acknowledge the authors' effort to adjust for this variable by including baseline lesion volume, time-to-treatment, and center effects as covariates in the mixed-effects models, the magnitude of the disparity between the NG101 (~729 mm³) and placebo (~400 mm³) groups remains a non-trivial confounder. It is plausible that the fluid dynamics and spontaneous resolution of edema in significantly larger lesions follow a different trajectory than in smaller lesions, a physical variance that statistical adjustment alone may not fully eliminate.

By explicitly acknowledging that the steeper decline in lesion volume may partly reflect early resolution of edema or hemorrhage, and by conceding that the MRI subcohort may not fully represent the randomized population, the authors have ensured that the conclusions are framed with the necessary scientific caution.

Reviewer #3

(Remarks to the Author)

Reviewer #4

(Remarks to the Author)

Thank you for the thorough and thoughtful revisions. I appreciate the authors' efforts in addressing all of the previous comments and clarifying the points raised. The responses are satisfactory, and the manuscript has improved significantly. I have no further concerns at this time.

Manuscript ID: NCOMMS-25-65782-T

Title: Anti-Nogo-A NG101 treatment induces changes in spinal cord micro- and macrostructure following spinal cord injury: A multicenter MRI study

We would like to thank the reviewers for their careful evaluation of our study and for their very relevant comments and suggestions, which have greatly improved the quality of this manuscript. We have addressed all of the comments and questions raised by the reviewers. Our responses below are in blue, the modified text that has been incorporated in the manuscript is shown below in *italic blue*.

Reviewer #1 (Remarks to the Author):

In the current manuscript, the authors investigated whether cross-sectional area (CSA) and Multiparameter Mapping (MPM)-derived indices can track NG101-induced structural effects and whether, based on preserved tissue bridges, patient stratification can potentially be improved in future trials. The authors analyzed longitudinal changes in spinal cord structure at and above the lesion site starting from screening (pre-treatment) to 1 month and then 6 months. Finally, the authors explored the combined utility of MRI and electrophysiological measures to identify responders and enhance the efficiency of future trials.

The authors have provided a strong manuscript. This reviewer would like to point out the following:

-There is a figure in the manuscript (unlabeled so it is hard to tell what number the figure is) which tracks “CSA” as y axis and the time since injury as x axis. This figure shows the preservation of CSA among cases treated with NG-101 compared to those treated with placebo. This corresponds to the lines 178-182 in the manuscript: “Above the level of lesion (C1-C2 level), at the screening visit, CSA was similar between treatment groups (NG101: n = 32, $69.86 \pm 6.46\text{mm}^2$; placebo: n=30, $66.25 \pm 6.03\text{mm}^2$; $p=0.065$), but declined significantly faster in placebo-treated participants ($-0.77\text{mm}^2/\text{month}$, 181 95%CI -1.10 to -0.44) than in those receiving NG101 ($-0.06\text{mm}^2/\text{month}$, 95% CI -0.82 to 0.70 ; $p=0.0020$). Therefore, it is fair to assume that these CSA measurements were performed at the C1-C2 level for all patients. As such, it might be helpful to then see what level the injury occurred for each patient so that meaningful comparisons may be performed. In other words, was the level of injury accounted for in these comparisons?

We thank the reviewer for pointing this out. All figures have now been properly labeled.

All linear mixed-effects models were corrected for the neurological level of injury (NLI). Cross-sectional Area (CSA) measurements were consistently performed at the C1-C2 level, which is above the lesion level for all patients. To clarify, we have added the following to the Methods section:

- *Fixed effects included treatment group (NG101 vs. placebo), time (days post baseline), their interaction (time x treatment), time to treatment initiation, age, sex, AIS, NLI, and the baseline value of the outcome measure. Center was modeled as a random intercept to account for site-related differences when model convergence was achieved; otherwise, it was included as a fixed effect. Subject-specific random slopes for time were included to capture within-subject changes over time.*

-Moreover, it would also be important to include snapshots of SCT -derived segmentations to determine if there were any artifact-related issues.

We have added supplementary material figure 8 showing SCT-derived CSA segmentations from randomly selected patients across all 8 centers included for CSA analysis. As described in the Results, 8 patients (7 NG101 and 1 placebo) were excluded from the CSA analysis due to artifacts.

-Along the same lines, the authors state in the discussion: “Our findings suggest that NG101 promotes tissue preservation and attenuates degeneration along sensorimotor pathways, with potential regenerative effects (even in a cohort including motor-complete participants) as demonstrated by slower progression of CSA atrophy.” Looking at the same figure, some subjects in both groups had a decrease in CSA from pre-treatment to 1 month and then an increase in the CSA by 6 months. While it is certainly evident from the figure that the CSA was preserved more in the patients receiving NG-101, the authors’ argument that there may be some “potential regenerative effects” attributable to NG-101 is challenged by this finding given that even some subjects in placebo demonstrate this effect.

Thank you for this important point. While some individual CSA trajectories show minor increases between 1 and 6 months in both NG101 and placebo participants, these fluctuations likely reflect biological variability—a phenomenon also reported in previous SCI studies (Freund et al., 2013, *Lancet Neurol.*; Ziegler et al., 2018, *Neurology*; Emmenegger et al., 2024, *Eur J Neurol.*) and do not negate the overall group-level finding of a significantly slower progression of CSA atrophy in the NG101 group. Notably, transient increases in cord area over time are a known phenomenon also observed in these studies. It remains speculative whether this reflects transient cord swelling due to early inflammatory processes, edema, or both. Technical factors such as slight variations in the analyzed vertebral level, cord angulation, or image quality (e.g. motion artifacts) can also contribute to within-subject variability in CSA estimates. Importantly, such effects are expected to occur across both treatment groups and therefore do not explain the overall group-level finding of a significantly slower progression of CSA atrophy in the NG101 group.

Reviewer #2 (Remarks to the Author):

This study was a multicenter, multinational, randomized, double-blind, placebo-controlled Phase 2b clinical trial—a framework that represents the gold standard for clinical investigation—to test whether anti-Nogo-A (NG101) treatment is associated with structural preservation/repair detectable by quantitative spinal cord MRI and whether MRI+SSEP can improve trial stratification. Each of these design elements serves a critical purpose in minimizing bias and ensuring the reliability of the results. The multicenter approach, involving specialized SCI centers across Germany, Switzerland, Spain, and the Czech Republic, enhances the diversity of the patient population and improves the generalizability of the findings beyond a single institution. Randomization, which allocated participants to either the NG101 or placebo group, is essential for preventing selection bias and creating groups that are, on average, comparable at the start of the study. The double-blinding procedure, where neither the participants nor the clinical staff knew who was receiving the active drug, is crucial for preventing performance and detection biases in both the administration of care and the assessment of outcomes. Finally, the inclusion of a placebo control group allows for the true effect of the drug to be isolated from the natural.

Major

1. Baseline imbalances and modeling choices

- Despite these considerable methodological strengths, the study suffers from a critical and ultimately unavoidable flaw that significantly impacts the interpretation of a subset of its findings. A detailed examination of the baseline characteristics reveals a statistically significant difference in a key prognostic variable between the two randomized groups. At the screening visit, before any intervention was administered, the group of participants randomized to receive NG101 had, on average, significantly larger lesion volumes than the group randomized to receive the placebo. The mean lesion volume in the NG101 group was reported as 729.1 mm³, compared to just 400.3 mm³ in the placebo group (p=0.011). This represents an 82% larger average lesion size in the active treatment arm at the outset of the study. This baseline imbalance is a major methodological weakness. While randomization is designed to produce comparable groups, by chance, it can sometimes fail to do so, particularly with smaller sample sizes. The failure to achieve comparability for a primary measure of injury severity introduces a powerful potential confounder. This confounder does not invalidate the entire study, but it does demand an extremely cautious and critical interpretation of any results related to changes in lesion volume over time.

We thank the reviewer for highlighting this important point. We agree that the baseline difference in lesion volume between treatment groups reflects an imbalance in a prognostically relevant variable. Participants were randomized according to predicted UEMS using the URP method, rather than imaging-derived measures, as MRI outcomes were defined as exploratory endpoints in the study protocol. Moreover, since the MRI cohort represents a subset of the full randomized population, this apparent imbalance could also reflect sampling effects specific to the imaging subset rather than true imbalance after randomization.

Due to initial space limitations, this crucial information had been omitted, but we confirm that all statistical models already included baseline lesion volume and other covariates (treatment group, time, time x treatment, age, sex, AIS, NLI). Importantly, all results remain robust when including lesion volume as a covariate. We have now clearly stated this in the methods section also added time to treatment and center as covariates to all models, as suggested by the reviewer:

- *Fixed effects included treatment group (NG101 vs. placebo), time (days post baseline), their interaction (time x treatment), time to treatment initiation, age, sex, AIS, NLI, and the baseline value of the outcome measure. Center was modeled as a random intercept to account for site-related differences when model convergence was achieved; otherwise, it was included as a fixed effect. Subject-specific random slopes for time were included to capture within-subject changes over time.*

Although some lesions were larger in the NG101 group at baseline, this group showed a faster apparent reduction in lesion volume over time. This likely reflects differences in MRI contrast (gray-scale intensity) rather than true tissue regrowth. The biological basis remains uncertain; however, in this study, faster volume reductions were primarily observed within the corticospinal tract (CST) and not in the dorsal columns, indicating that the treatment did not influence Wallerian degeneration, where sprouting is not possible, as the parent neurons are located below the injury. Importantly, our MRI measurements were obtained at and above the lesion, where sprouting and local remodeling can occur. This observation aligns with our previous macaque work (Freund et al., 2006; 2007), where we found collateral sprouting of corticospinal fibers in the vicinity of the lesion, precisely in regions showing faster CST volume reductions. Taken together, these findings suggest that the lesion volume shrinkage observed in NG101-treated participants is unlikely to be a confounding effect of baseline imbalance but rather may reflect treatment-driven remodeling in descending tracts.

- The implications of this baseline difference are profound. A primary finding reported by Farner et al. is a "faster monthly reduction in lesion volume" in the NG101 group, which is interpreted as a positive therapeutic effect indicative of drug-induced tissue remodeling or resolution of edema. However, this interpretation fails to adequately account for the initial disparity. In the acute and subacute phases of SCI, a substantial portion of the hyperintense signal that constitutes the "lesion volume" on a T2-weighted MRI scan is not necrotic tissue but rather vasogenic edema and hemorrhage^{1,2}. This fluid component resolves naturally over the first weeks and months post-injury, leading to a spontaneous reduction in the overall measured lesion volume. A larger initial lesion, particularly one with a greater inflammatory and edematous component, possesses a mathematically greater absolute volume to lose during this natural resolution process. Consequently, the observed "faster reduction" in the NG101 group may be, in whole or in significant part, an artifact of this baseline difference rather than a direct consequence of a regenerative drug effect. The authors' attribution of this finding to "axonal sprouting" or "glial reorganization" is highly speculative and potentially invalid due to this unaddressed confound. This forces a re-evaluation of the paper's evidence, shifting the burden of proof for a biological effect of NG101 away from the confounded focal lesion data and onto the analysis of remote biomarkers that are geographically distinct from the primary injury.

We agree with the reviewer that interpretation of lesion volume reduction as evidence of axonal sprouting or glial reorganization is confounded by baseline differences and the natural evolution of edema in the acute and subacute phases (see also Emmenegger et al, *Eur. J. Neurol*, 2024). This is now explicitly acknowledged in the revised manuscript in the discussion:

- *The steeper decline in lesion volume in the NG101 group may partly reflect early resolution of edema or hemorrhage, but could also involve axonal sprouting, formation of new connections, or glial reorganization in the perilesional area.¹³ Importantly, the faster apparent volume reduction was primarily observed within the CST, whereas the DC did not show a comparable decline. This tract-specific pattern suggests that the observed effect is not due to generalized Wallerian degeneration, which would also affect the DC, but may instead reflect remodeling or plasticity within descending motor pathways. Preclinical evidence supports that Nogo-A inhibition promotes axonal plasticity and rewiring around the lesion.² Thus, the lesion-volume trajectory may reflect a combination of spontaneous fluid resolution and potential drug-induced tissue effects, although the relative contribution of each cannot be definitively determined from the current data.*

However, even after accounting for these factors, evidence of treatment-related effects remains in metrics that are less influenced by focal lesion confounds. Specifically, changes in remote myelin-sensitive biomarkers (MTsat) along the corticospinal tract (CST) and dorsal columns (DC) at C1-C2, as well as cervical spinal cord area (CSA) at C1-C2, which are geographically distant from the primary lesion, support a potential biological effect of NG101. These findings, combined with statistical adjustment for baseline volume, allow us to cautiously interpret treatment effects beyond the natural evolution of edema, while maintaining appropriate caution regarding speculative mechanistic claims.

- The analysis of structural changes at the epicenter of the injury presents a complex and ultimately ambiguous picture. The data show a significantly faster rate of decline in both lesion volume (-19.9 mm³/month in the NG101 group vs. -2.8 mm³/month in the placebo group; p=0.005) and lesion length in the NG101-treated participants.¹ As established in the preceding section; these findings must be

interpreted with extreme caution. Due to the significant baseline imbalance, where the NG101 group started with much larger lesions, these results cannot be unambiguously attributed to a therapeutic effect. It is equally plausible that they reflect the natural history of the resolution of larger initial injuries. Therefore, the evidence for a drug-induced effect on the primary lesion itself is weak and confounded.

We thank the reviewer for this important point and fully agree that structural changes at the injury epicenter must be interpreted with caution. The NG101 group started with significantly larger baseline lesions, and although all analyses were adjusted for baseline lesion volume, we cannot completely exclude spontaneous fluid resolution as addressed in the answers above.

- Consider adding fixed covariates (age, AIS grade, NLI, time-to-treatment, site) and random slopes for time to better capture inter-individual trajectories and site heterogeneity.

In response to the reviewer's suggestion, we re-ran all longitudinal mixed-effects models to explicitly include time to treatment initiation and center as additional covariates and stated specifically in the methods section which covariates have been included in the mixed-effect models. All models already included treatment group (NG101 vs. placebo), time (days post-baseline), and their interaction (time x treatment) as primary fixed effects, as well as age, sex, AIS grade, neurological level of injury (NLI), and the baseline value of the respective outcome measure. Center effects were modeled as random intercepts to account for site-level variability whenever convergence permitted; in cases of convergence or singularity issues, center was included as a fixed effect. Subject-specific random slopes for time were also explored. The results from these updated models were fully consistent with the original analyses, with no changes in statistical significance or interpretation. The Methods section has been updated to reflect these changes.

- *Fixed effects included treatment group (NG101 vs. placebo), time (days post baseline), their interaction (time x treatment), time to treatment initiation, age, sex, AIS, NLI, and the baseline value of the outcome measure. Center was modeled as a random intercept to account for site-related differences when model convergence was achieved; otherwise, it was included as a fixed effect. Subject-specific random slopes for time were included to capture within-subject changes over time.*

- The MRI subset exhibits baseline imbalances that warrant additional adjustment and sensitivity analyses. Rather than “topping up” enrollment post hoc to erase these differences—which is rarely appropriate after randomization and may introduce temporal/site biases—please

(i) predefine and report ANCOVA/mixed-effects models that include key baseline MRI covariates (e.g., lesion volume/length/width, TB, time from injury),

To address baseline imbalances in the MRI subset, we included baseline values of the respective outcome as a covariate in the main models. Additional lesion metrics (lesion length, lesion width, tissue bridges) at baseline were added as covariates for sensitivity analysis for lesion volume but it did not change the overall significance of the model (see R.Table 1 below).

(ii) adjust or stratify by site/scanner where feasible, and

To account for site heterogeneity, center was included as a random intercept whenever supported by model convergence; otherwise, center was added as a fixed effect. This approach allowed us to consistently capture between-center variability without singular fits.

(iii) provide sensitivity analyses, consider adding fixed covariates (age, AIS grade, NLI, time-to-treatment, site) and random slopes for time to better capture inter-individual trajectories and site heterogeneity.

As suggested by the reviewer, we performed sensitivity analyses to model lesion volume over time and assess the robustness of our findings to alternative model specifications and additional baseline covariates. We began with a minimal covariates model only including as fixed effects baseline lesion volume, a random intercept for center, and a patient-specific random slope for time.

We then looked at the full covariates model, including as fixed effects baseline lesion volume, sex, time-to-treatment, age, NLI, AIS, a random intercept for center, and a patient-specific random slope for time

Tissue bridges, lesion length, or lesion width at baseline were subsequently added as additional covariates. For these extended models, center was included as a fixed effect, as models did not converge when modeled as a random intercept.

Across all models, the estimated slopes for NG101 versus placebo remained consistent, indicating that our conclusions regarding treatment-related lesion volume change are robust to these different model specifications and adjustments for baseline MRI features.

R. Table 1. Sensitivity Analysis Assessing Lesion Volume Across Model Specifications

Model	Group	Estimated Change / Effect (β, mm³/month)	95% CI (mm³/month)	p-value	Covariates
Minimal Covariates Model	NG101	-58.56	-83.59 to -33.52	0.009	Fixed effects: baseline lesion volume, random intercept for center, random slope for time
	Placebo	-8.00	-36.65 to 20.64		
Full Covariates Model	NG101	-52.14	-78.96 to -25.33	0.012	Fixed effects: baseline lesion volume, sex, time to treatment, age, NLI, AIS, random intercept for
	Placebo	-2.25	-33.46 to 28.96		

					center, random slope for time
Center as fixed effect	NG101	-52.86	-80.04 to -25.68	0.009	Fixed effects: baseline lesion volume, sex , time to treatment, age, NLI, AIS, center, random slope for time
	Placebo	0.67	-31.26 to 32.60		
+ Tissue bridges at baseline	NG101	-52.86	-80.04 to -25.68	0.009	Fixed effects: baseline tissue bridges, baseline lesion volume, sex , time to treatment, age, NLI, AIS, center, random slope for time
	Placebo	0.67	-31.26 to 32.60		
+ Lesion Length at baseline	NG101	-52.72	-79.88 to -25.57	0.009	Fixed effects: baseline lesion length, baseline lesion volume, sex , time to treatment, age, NLI, AIS, center, random slope for time
	Placebo	0.41	-31.57 to 32.38		
+ Lesion Width at baseline	NG101	-54.83	-81.93 to -27.74	0.009	Fixed effects: baseline lesion width, baseline lesion volume, sex , time to treatment, age, NLI, AIS, center, random slope for time
	Placebo	-1.87	-33.80 to 30.06		

Given the mid-trial change in allocation ratio (initially 1:1, later 3:1 to achieve ~2:1 overall), please also discuss whether this operational change could have contributed to the observed baseline imbalances.

We thank the reviewer for this comment. While the mid-trial change in allocation from 1:1 to 3:1 could theoretically affect baseline balance, randomization was maintained within each scheme and stratified by URP nodes. Participants were assigned based on predicted 6 month UEMS, not imaging measures, and all analyses were adjusted for baseline prognostic variables to account for any residual imbalances.

2. Unacknowledged Limitations

Beyond the authors' own assessment, a deeper critique reveals other critical issues that temper the study's conclusions.

- The Overarching Confound of Baseline Lesion Volume: This must be revisited and framed as the most significant unacknowledged weakness of the study. The failure of randomization to create comparable groups on this key prognostic variable fundamentally undermines the conclusions that can be drawn about any treatment effects on the focal lesion itself.

We thank the reviewer for this valuable comment and agree that baseline lesion volume represents a key prognostic variable that warrants explicit acknowledgment. MRI data were available only for a subset of participants, and within this imaging cohort, baseline lesion volumes showed some imbalance between treatment groups. Because MRI data were not obtained for the entire randomized NISCI cohort, we cannot determine whether this reflects a true imbalance after randomization or a sampling effect specific to the imaging subset.

To address this, we have revised the limitations section as follows:

- Baseline imbalances in lesion volume within the MRI subcohort suggest that the analyzed imaging sample may not fully represent the randomized NISCI population. Although statistical adjustments were made to account for these differences, such imbalances could still influence treatment effect estimates. This limitation should be considered when interpreting the results, and future studies with larger imaging datasets will be needed to confirm these findings.

We believe this clarification more accurately reflects the available data and appropriately qualifies the strength of the conclusions regarding treatment effects on lesion volume.

- The Post-Hoc Nature of the Stratification Analysis: It is crucial to stress that the powerful stratification strategy was developed through a retrospective, post-hoc analysis of the trial data. This is a valid and extremely useful approach for generating new hypotheses. However, the results do not carry the same weight of evidence as a prospectively defined analysis. The true test of this strategy will require it to be applied prospectively in a new clinical trial where the "responder" subgroup is the primary population of interest from the outset. Moreover, similar study has reported this³.

We added a sentence to address the post-hoc stratification to the limitations:

- *The stratification strategy was developed retrospectively in post-hoc analyses. While this approach has been validated in previous studies¹¹ and can provide useful estimates, its retrospective nature limits the interpretation of prospective data and carries an inherent risk of type I error. These findings should therefore be confirmed in a future clinical trial designed*

prospectively with the “responder” subgroup as the primary population of interest. To mitigate potential bias and strengthen the robustness of our findings, non-parametric bootstrapping (10,000 resamples) was applied to effect sizes and mean differences. Independent, prospectively designed cohorts will be required to confirm the reproducibility of these results.

Minor:

1. Imaging protocol clarity (potential inconsistency)

- The Results/Participants section attributes limited MPM availability to “scanner field strength limitations,” yet Methods state all scans were acquired on 3 T systems. This reads as contradictory. Please reconcile and detail per-site vendors/sequences, harmonization steps, and how site effects were modeled.

We changed the sentence in the methods to accurately state that there were also data from 1.5 T scanners included in the study:

- *All scans were acquired on 1.5 or 3 Tesla MRI systems. Scanner details are provided in Supplementary Material Table 2.*

Additionally we have added supplementary material table 2 showing the scanner types and field strength for each center.

2. Multiplicity and statistical transparency

- Numerous outcomes are tested (lesion volume/length/width, CST/DC overlap, CSA and widths, MTsat in CST/DC, several stratifications). No correction for multiplicity is described. Please prespecify primary imaging endpoints and apply a multiplicity control (e.g., FDR). Report effect sizes with 95% CIs consistently for all primary/secondary outcomes.

As specified in the NISCI study protocol, this was an exploratory imaging analysis rather than a confirmatory clinical trial. Therefore, no formal correction for multiplicity was applied (Perneger, *BMJ*, 1998; Fremantle, *BMJ*, 2001; Moyé, *Contemp Clin Trials*. 2015). The imaging measures (e.g., lesion volume, tract overlap, CSA, MTsat metrics, stratifications) were pre-planned exploratory outcomes intended to comprehensively characterize potential biomarkers of treatment response. Linear mixed-effects model outcomes and effect sizes are reported with 95% confidence intervals.

- *All results are presented with effect estimates, p-values, and 95% confidence intervals.*

3. Stratification & “post-hoc power”

- The improvement in effect sizes using TB \geq 1.0 mm plus preserved SSEP is promising, but appears post-hoc and uses observed effects to back-calculate required sample sizes. This can be optimistic.

Please

(i) clarify whether these stratifications were pre-specified,

The TB cutoff of \geq 1.0 mm was derived from a multicenter cohort (Pfyffer et al., *Lancet Neurol.*, 2024), where it was determined using a URPC-TREE model of the 3-month UEMS outcome. SSEP preservation

was assessed at screening as either preserved or abolished, reflecting the pre-existing functional integrity of the somatosensory pathways as described by Scheuren et al., *Brain*, 2025.

(ii) provide bias-corrected effect sizes (e.g., bootstrap, nested CV), and

We performed non-parametric bootstrapping (10,000 resamples) within each stratified subgroup to derive bias-corrected mean differences between NG101 and placebo. For each outcome ($\Delta UEMS$, $\Delta SCIM$ self-care), we now report the effect sizes as well as the observed mean difference together with its bootstrapped 95% confidence interval in the results. We addressed this in the methods as well and added a sentence:

- *To provide bias-corrected effect estimates, we applied non-parametric bootstrapping (10,000 resamples) to calculate both effect sizes and mean differences between NG101 and placebo, together with their 95% percentile confidence intervals.*

(iii) avoid “post-hoc power”; instead report CIs and, if appropriate, prospective simulations for future trials.

We have added 95% confidence intervals for all reported results throughout the manuscript.

4. Microstructural metrics

- Please add quality control details for MTsat (e.g., motion rejection thresholds, B1 correction), test–retest reliability references for cord MTsat, and discuss partial-volume handling at C1/C2. Also, clarify whether CST/DC ROIs were lesion-unaffected at C1/C2 across all subjects.

All CSA measurements were obtained at the C1/C2 level, which was above the lesion site in all patients. To address the reviewer’s point, we have also added a sentence in the Methods section clarifying the quality control procedures.

- *Scanner harmonization was ensured by following a standardized multicenter protocol, previously validated,^{10,22} which demonstrated consistent parameter estimates in the cervical spinal cord across seven scanners, with intra- and inter-site coefficients of variation ranging from 2.5–12% for MT, R1,*

and PD, and 1.1–4.0% for morphometric measures. These images were used to compute quantitative maps of MTsat via the hMRI toolbox²⁵ embedded in SPM12 (UCL, London, UK), which applies corrections based on separately acquired B1⁺ and B1⁻ maps to account for transmit and receive field inhomogeneities. Due to poor image quality at C3 across the imaged cohort, analysis focused on the C1-C2 levels, which was above the lesion level for all participants.

- To mitigate partial volume effects, cross-sectional area (CSA) and tract-specific metrics were extracted using SCT's atlas-based weighted masks and maximum a posteriori estimation, which account for mixed tissue contributions at voxel boundaries

5. Manual segmentations & reproducibility

- Lesion segmentation was manual with blinded operators. Please report inter-/intra-rater reliability and make annotation guidelines/code available.

We thank the reviewer for highlighting the importance of segmentation reproducibility. All lesion segmentations were performed manually by a rater blinded to timepoint and participant identity. To assess intra-rater reliability, a subset of 15 randomly selected cases was re-segmented after 19 days, and agreement was quantified using both the coefficient of variation (COV) and the intraclass correlation coefficient (ICC).

Across lesion measures, reproducibility was excellent, with a mean intra-rater COV of 5.3% (range: 3.8–8.4%) and ICC values ranging from 0.96 to 0.999. These results are in line with previously reported intra-rater variability for spinal cord lesion segmentation (Pfyffer et al., *Neurology*, 2019; Huber et al., *Annals of Neurology*, 2017). Supplementary Material Figure 7 details the anatomical landmarks used for manual delineation.

6. Define specialized terms earlier and correct terminology

- Several domain-specific measures appear in the Results without prior plain-language definitions, which may hinder readability for non-clinician readers (e.g., UEMS, SCIM, MTsat). While UEMS and the SCIM self-care subscore are described in the Methods, these definitions come after their first substantive use in the text. Please define each acronym on first mention in the Abstract/Results and consider adding a brief glossary.

All acronyms have now been defined at first mention in the Abstract and Results.

7. Concomitant treatments beyond NG101

- The Intervention/Methods section clearly describes the blinded intrathecal dosing schedule of NG101 versus saline placebo, but the manuscript does not specify permitted/prohibited concomitant therapies or standardization of rehabilitation/acute surgical care across sites within this MRI study. Clarifying these co-interventions is essential to assess potential confounding of imaging and clinical outcomes.

- Please add

(i) a concise description of allowed and disallowed concomitant treatments (e.g., timing of decompression/fixation, corticosteroid use, rehabilitation intensity/dose),

We thank the reviewer for raising this point. Patients underwent standard-of-care decompression and stabilization procedures as clinically indicated, with timing and approach determined by local treating

physicians according to standard guidelines. All patients received standard rehabilitation at their respective sites (Weidner et al., *Lancet Neurol.*, 2024). We added the full inclusion and exclusion criteria in the supplementary material and added the following sentence to the methods section:

- *All participants received standard-of-care acute surgical management and rehabilitation at their respective centers.⁵ Inclusion and exclusion criteria are described in the Supplementary Material.*

(ii) whether these were balanced between groups, and

Although concomitant therapies were not actively balanced through randomization, standardized documentation of rehabilitation intensity and content (Mapping of Rehabilitative Training, MART) indicated comparable rehabilitative interventions between treatment groups within the first 84 days post baseline (unpublished observation).

(iii) whether any such variables were adjusted for in the longitudinal models.

Since concomitant therapies appeared balanced and their inclusion as separate covariates would have substantially increased model complexity, they were not included in the longitudinal models.

Reference:

1. Margetis, K., Das, J. M. & Emmady, P. D. Spinal Cord Injuries. in StatPearls (StatPearls Publishing, Treasure Island (FL), 2025).
2. Zhang, Y. et al. Acute spinal cord injury: Pathophysiology and pharmacological intervention (Review). *Molecular Medicine Reports* 23, 1–18 (2021).
3. Pfyffer, D. et al. Prognostic value of tissue bridges in cervical spinal cord injury: a longitudinal, multicentre, retrospective cohort study. *The Lancet Neurology* 23, 816–825 (2024).

Reviewer #3 (Remarks to the Author):

We thank the Reviewer for their contribution and appreciate the time and effort dedicated to providing feedback on our manuscript.

Reviewer #4 (Remarks to the Author):

Key Results

This multicenter phase 2b study investigates the effects of anti-Nogo-A antibody NG101 in acute cervical

SCI using quantitative MRI and electrophysiology. The authors report that NG101 treatment attenuates lesion volume growth, preserves spinal cord cross-sectional area (CSA), and slows decline in magnetization transfer saturation (MTsat) in corticospinal and dorsal column tracts. Integration of MRI biomarkers with electrophysiology (SSEP) enhanced stratification and improved effect sizes for functional outcomes (UEMS, SCIM self-care). Findings suggest NG101 may mitigate macro- and microstructural degeneration and that multimodal stratification can optimize trial design.

Validity

The study is robust in design and employs rigorous imaging and electrophysiological protocols across multiple centers. Statistical methods (linear mixed-effects modeling, effect size estimation, sample size projection) are appropriate.

However, several limitations and potential sources of bias require clearer acknowledgment:

- Attrition and missing data: Imaging subsets are substantially smaller than the full cohort (e.g., CSA/MTsat analyses included only ~62 participants). If dropouts or exclusions disproportionately involved patients with more severe injuries, results could be biased toward “healthier” participants.

We thank the reviewer for raising this important point. Indeed, the imaging subsets are smaller than the full clinical cohort. The main reasons for missing MRI data were (i) no consent given for MRI analysis, (ii) unavailability of multi-parameter mapping (MPM) sequences at certain centers used for CSA analysis, and (iii) logistical or technical issues, including motion or metal artefacts, scanner availability, and scheduling constraints, that prevented participants from completing all imaging sessions. Importantly, dropouts were not systematically related to injury severity. As described in the Results section, clinical baseline measures shown for each subcohort (AIS grade, demographics) did not significantly differ from those of participants without imaging data, suggesting that the imaging subset was not biased toward “healthier” patients. Furthermore, AIS distributions were balanced between treatment arms within each subcohort. We have clarified these points in the revised manuscript by adding Figure 1, showing a flowchart summarizing participant screening.

- Stratification: The refined stratification (TB \geq 1 mm + preserved SSEP) is highly informative but was conducted retrospectively, in small subgroups. This increases the risk of type I error and inflated effect sizes.

We agree and have addressed this more fully in the limitations section:

- *The stratification strategy was developed retrospectively in post-hoc analyses. While this approach has been validated in previous studies¹¹ and can provide useful estimates, its retrospective nature limits the interpretation of prospective data and carries an inherent risk of type I error. These findings should therefore be confirmed in a future clinical trial designed prospectively with the “responder” subgroup as the primary population of interest. To mitigate potential bias and strengthen the robustness of our findings, non-parametric bootstrapping (10,000 resamples) was applied to effect sizes and mean differences. Independent, prospectively designed cohorts will be required to confirm the reproducibility of these results.*

- Imaging acquisition: Advanced MPM protocols were only implemented at select sites, and scanner variability may introduce noise despite harmonization efforts.

We added a sentence to the limitations to address this:

- *This was a multi-site longitudinal study, and scanner-related variability may have introduced additional noise. Although a harmonized MPM protocol and established correction methods were applied,^{10,22} residual site-related effects cannot be entirely excluded, and analyses were restricted to C1–C2 due to poor image quality at C3.*

- Structure–function disconnect: Structural preservation was observed even in motor-complete participants without functional gains, raising the possibility that some imaging changes reflect subclinical processes (e.g., edema resolution, gliosis) rather than functional regeneration.

We agree that structural preservation at the lesion epicenter may in part reflect subclinical processes such as edema resolution or gliosis rather than true regeneration (Ellingson et al., *World Neurosurg.*, 2014, Noristani, Perrin, *Neural Regen Res.*, 2019). However, the pattern of changes observed in remote biomarkers, including corticospinal tract and dorsal column MTsat at C1–C2, as well as cervical spinal cord area (CSA), provides complementary evidence for a treatment effect beyond the lesion site. These regions are anatomically distinct from the primary injury and less likely to be influenced by local edema or gliotic changes. While imaging findings alone cannot establish functional recovery, they suggest that NG101 may mitigate secondary degeneration and preserve structural integrity in remote pathways, supporting a potential neuroprotective or regenerative effect.

These issues do not undermine the core findings but should temper interpretation and be more explicitly discussed.

Significance

This work is highly significant for SCI research:

- Demonstrates, for the first time, that anti–Nogo-A treatment produces measurable in vivo macro- and microstructural changes.
- Supports MRI and electrophysiology as biomarkers to improve trial sensitivity, reduce sample sizes, and refine inclusion criteria.
- Findings have broad translational relevance for neuroregeneration trials beyond SCI.

Potential limitation: the absence of clear clinical improvement in motor-complete participants may temper enthusiasm for immediate clinical impact.

Data and Methodology

Imaging and electrophysiology protocols are state-of-the-art, with careful quality control and blinded analyses. MRI biomarkers (CSA, MTsat) are well validated for multicenter studies.

However, the paper should more explicitly address:

- The impact of missing data and variable sample sizes across analyses.

In our study, sample sizes varied across analyses due to missing or excluded data, primarily arising from withdrawals, lack of consent for MRI data analysis, technical limitations (e.g., scanner availability, protocol differences), or exclusion of images with inadequate quality. To address this, we consistently analyzed the maximum number of participants available for each dataset to optimize statistical power. Importantly, these sources of missing data were unrelated to treatment allocation, and baseline demographic and injury characteristics remained well balanced between NG101 and placebo across all analyzed subgroups, as described in the Results section. Thus, while variable sample sizes inevitably reduced the power of stratified analyses and may affect direct comparability across analyses, the risk of systematic bias was minimized. We have revised the manuscript to make this rationale more explicit in the Limitations section.

- *Although the overall cohort was relatively large for a spinal cord imaging study, stratified subgroups were small, and sample sizes varied across analyses due to missing or excluded data, which may affect comparability.*

- How harmonization across scanners was achieved and validated.

We added a sentence to the methods to explain the harmonization process across scanners:

- *Scanner harmonization was ensured by following a standardized multicenter protocol, previously validated,^{10,22} which demonstrated consistent parameter estimates in the cervical spinal cord across seven scanners, with intra- and inter-site coefficients of variation ranging from 2.5–12% for MT, R1, and PD, and 1.1–4.0% for morphometric measures.*

- The rationale for selecting TB ≥ 1 mm as the cut-off, and whether sensitivity analyses using other thresholds were tested.

The TB cutoff of ≥ 1.0 mm was derived from a multicenter cohort (Pfyffer et al., *Lancet Neurol.*, 2024), where it was determined using a URP model of the 3 month UEMS outcome. In the present study, we applied this predefined threshold and did not perform sensitivity analyses with alternative cutoffs.

Analytical Approach

- Linear mixed-effects models and effect size/sample size estimations are appropriate.
- Post-hoc subgroup analyses are informative, though risk of type I error inflation should be acknowledged.

We added a sentence to the limitations to address this:

- *The stratification strategy was developed retrospectively in post-hoc analyses. While this approach has been validated in previous studies¹¹ and can provide useful estimates, its retrospective nature limits the interpretation of prospective data and carries an inherent risk of type I error. These findings should therefore be confirmed in a future clinical trial designed prospectively with the “responder” subgroup as the primary population of interest. To mitigate potential bias and strengthen the robustness of our findings, non-parametric bootstrapping (10,000 resamples) was applied to effect sizes and mean differences. Independent, prospectively designed cohorts will be required to confirm the reproducibility of these results.*

- Consider sensitivity analyses to confirm robustness of treatment effects in stratified groups.

We performed sensitivity analyses using non-parametric bootstrapping (10,000 resamples) within each stratified subgroup to derive bias-corrected estimates of the treatment effect. For each outcome (Δ UEMS, Δ SCIM self-care), we now report the effect sizes as well as the observed mean difference together with its bootstrapped 95% confidence interval. This approach provides a more robust estimate of the uncertainty around the treatment effect and directly addresses potential optimism bias. The methods section has been updated to describe this procedure:

- *To provide bias-corrected effect estimates, we additionally applied non-parametric bootstrapping (10,000 resamples) to calculate both effect sizes and mean differences between NG101 and placebo, together with their 95% percentile confidence intervals.*

Suggested Improvements

- Include a CONSORT-style flow diagram summarizing patient screening, randomization, exclusions, and follow-up across imaging and clinical analyses to enhance transparency.

Thank you for the suggestion. We have now included a figure summarizing patient screening, randomization, and exclusions across imaging and clinical analyses as Figure 1.

- Clarify that stratification was based on baseline TB/SSEP status, not longitudinal changes, and that functional improvements were concentrated in those with preserved tissue and electrophysiology.

We revised the sentence in the discussion to specify this more clearly:

- *While NG101 treatment led to improvements in UEMS and SCIM self-care scores among motor-incomplete participants, stratification based on combined structural (TB) and functional (tibial SSEP or C8 dSEEPS) biomarkers measured at the screening visit revealed much larger treatment effects and dramatically reduced the required sample sizes.*

- Explicitly state that non-responders (TB <1 mm or absent SSEP) showed no treatment effect, to avoid misinterpretation.

We added this sentence to the discussion:

- *In contrast, participants with TB < 1 mm or abolished SSEP showed no treatment effect, and TB > 1 mm alone was insufficient to confer a treatment benefit without preserved SSEP.*

- Provide greater discussion of limitations: missing data, subgroup size, baseline imbalances, and potential scanner-related variability.

We agree with this and have therefore extended the limitations section to address issues raised by the reviewer, including scanner-related variability, subgroup sizes, baseline imbalances, missing data, and the retrospective nature of the stratification, with full details provided in the text below, as written in the limitations section of the manuscript:

- *Several limitations warrant consideration. This was a multi-site longitudinal study, and scanner-related variability may have introduced additional noise. Although a harmonized MPM protocol and established correction methods were applied,^{10,22} residual site-related effects cannot be entirely excluded, and analyses were restricted to C1-C2 due to poor image quality at C3. While MTsat is sensitive to myelin, it is not entirely specific and may also reflect other microstructural processes.²³ Although the overall cohort was relatively large for a spinal cord imaging study, stratified subgroups were small, and sample sizes varied across analyses due to missing or excluded data, which may affect comparability. Baseline imbalances in lesion volume within the MRI subcohort suggest that the analyzed imaging sample may not fully represent the randomized NISCI population. Although statistical adjustments were made to account for these differences, such imbalances could still influence treatment effect estimates. This limitation should be considered when interpreting the results, and future studies with larger imaging datasets will be needed to confirm these findings. The stratification strategy was developed retrospectively in post-hoc analyses. While this approach has been validated in previous studies¹¹ and can provide useful estimates, its retrospective nature limits the interpretation of prospective data and carries an inherent risk of type I error. These findings should therefore be confirmed in a future clinical trial designed prospectively with the “responder” subgroup as the primary population of interest. To mitigate potential bias and strengthen the robustness of our findings, non-parametric bootstrapping (10,000 resamples) was applied to effect sizes and mean differences. Independent, prospectively designed cohorts will be required to confirm the reproducibility of these results.*

- Where possible, report correlations between structural biomarkers (e.g., TB, CSA, MTsat) and functional recovery (UEMS/SCIM) to strengthen biological interpretation.

We examined associations between structural MRI biomarkers (e.g., lesion metrics, TB, CSA, MTsat) and functional recovery (UEMS and SCIM self-care) using Pearson correlation coefficients and confirmed previous reports (Pfyffer et al., *Neurology*, 2019; Huber et al., *Annals of Neurol.*, 2017). Both baseline values and 6-month changes were considered, with significant correlations presented in Supplementary

Material Table 3. As this analysis was not the primary focus and extends beyond the main scope of the current paper, we included it in the supplementary material; however, it can be incorporated into the main text if the reviewer requests. This approach is also described in the methods section.

- *To assess structure-function relationships, Pearson correlation coefficients were calculated between baseline and changes ($\Delta[6\text{months} - \text{baseline}]$) in MRI biomarkers (e.g., lesion metrics, CSA, MTsat) and 6-month functional recovery measures (UEMS and SCIM self-care). Significant correlations are reported in Supplementary Material Table 3.*

Clarity and Context

- Manuscript is well-written, with clear figures and logical flow.
 - Contextualization with prior work is strong (e.g., citing MRI biomarkers, Nogo-A preclinical data).
 - Consider shortening certain result subsections for readability, as the manuscript is data-dense.
-
- The authors should emphasize the conceptual advance that multimodal stratification is superior to conventional AIS-based grouping more prominently in the Discussion.

We thank the reviewer for this valuable suggestion. We have revised the Discussion to emphasize more clearly the conceptual advance of our work:

- *This multimodal stratification identified a responder subgroup that conventional clinical-based classification would have overlooked, highlighting that combined imaging and electrophysiological markers provide a more precise and clinically meaningful framework for predicting treatment response than clinical grouping alone. Our findings further demonstrate that it is the combination of structural integrity and functional preservation that offers the most powerful and reliable stratification tool. While MRI alone lacked sufficient predictive precision, electrophysiological stratification based on preserved tibial SSEP proved especially valuable.¹¹ Importantly, the combination of both electrophysiological and MRI markers further improved stratification. This concept aligns with previous work emphasizing that an integrated structural–functional perspective is essential to capture neural plasticity and repair mechanisms after spinal cord injury.²¹ Such a multimodal approach may not only serve to increase statistical power and treatment effect but also to enhance trial efficiency by reducing required sample sizes. This may eventually enable more targeted and effective clinical trials in acute SCI.*

References

The references are comprehensive and appropriate, with inclusion of both preclinical and clinical studies. Some citations to under-review work should be updated if possible.

Reviewer Expertise:

I am experienced in clinical trial design, neurorehabilitation, and imaging biomarkers in SCI. I defer detailed assessment of raw MRI sequence implementation to technical experts.

Overall Assessment:

This is a technically rigorous and conceptually innovative study. It advances the field by demonstrating that NG101 induces measurable structural preservation and that multimodal biomarker

stratification can enhance trial sensitivity. With clearer acknowledgment of limitations, addition of a CONSORT-style flow diagram, and stronger emphasis on the stratification findings, the manuscript will make a valuable contribution to Nature Communications.

Anti-Nogo-A NG101 treatment induces changes in spinal cord micro- and macrostructure following spinal cord injury

We would like to express our sincere gratitude to all the reviewers for their time, constructive feedback, and positive evaluation of our revised manuscript. Their insights during the review process have been instrumental in significantly improving the rigor, transparency, and quality of our work.

Reviewer #1 (Remarks to the Author): *All previous comments have been adequately addressed.*

We thank the reviewer for their thorough evaluation of our initial manuscript and their confirmation that our revisions have adequately addressed all previous comments.

Reviewer #2 (Remarks to the Author): *The authors have submitted a substantially revised manuscript that addresses the majority of the concerns raised in the previous round of review. I commend the detailed and thoughtful nature of the rebuttal. The inclusion of the flow diagram significantly enhances the transparency of participant selection and exclusion across the clinical and imaging sub-cohorts. Furthermore, the methodological shift from post-hoc power calculations to non-parametric bootstrapping for effect size estimation represents a distinct improvement, providing more robust and bias-corrected confidence intervals. The additional sensitivity analyses presented in the rebuttal, which tested the robustness of the findings against various model specifications and covariates, are also well-received. However, I retain a degree of skepticism regarding the baseline imbalance in lesion volume. While I acknowledge the authors' effort to adjust for this variable by including baseline lesion volume, time-to-treatment, and center effects as covariates in the mixed-effects models, the magnitude of the disparity between the NG101 (~729 mm³) and placebo (~400 mm³) groups remains a non-trivial confounder. It is plausible that the fluid dynamics and spontaneous resolution of edema in significantly larger lesions follow a different trajectory than in smaller lesions, a physical variance that statistical adjustment alone may not fully eliminate. By explicitly acknowledging that the steeper decline in lesion volume may partly reflect early resolution of edema or hemorrhage, and by conceding that the MRI subcohort may not fully represent the randomized population, the authors have ensured that the conclusions are framed with the necessary scientific caution.*

We sincerely thank the reviewer for their positive feedback and for recognizing the value of the additions to the manuscript. We completely understand and respect the reviewer's residual caution regarding the baseline imbalance in lesion volume. We are glad the reviewer agrees that our explicit discussion of these confounding factors in the revised manuscript provides the necessary transparency, ensuring that readers interpret these specific findings with appropriate scientific caution. We thank the reviewer again for their critical and highly constructive engagement with our study.

Reviewer #3 (Remarks to the Author): *I co-reviewed this manuscript with one of the reviewers who provided the listed reports. This is part of the Nature Communications initiative to facilitate training in peer review and to provide appropriate recognition for Early Career Researchers who co-review manuscripts.*

We thank the co-reviewer for their valuable contribution to the peer review process.

Reviewer #4 (Remarks to the Author): *Thank you for the thorough and thoughtful revisions. I appreciate the authors' efforts in addressing all of the previous comments and clarifying the*

points raised. The responses are satisfactory, and the manuscript has improved significantly. I have no further concerns at this time.

We are very grateful to the reviewer for their positive assessment of our revised manuscript and their confirmation that our responses are satisfactory.